# A Geometric Framework for Understanding Memorization in Generative Models

**Brendan Leigh Ross, Hamidreza Kamkari, Tongzi Wu, Rasa Hosseinzadeh,**
**Zhaoyan Liu, George Stein, Jesse C. Cresswell, Gabriel Loaiza-Ganem**
Layer 6 AI
{brendan,hamid,tongzi,rasa,zhaoyan,george,jesse,gabriel}@layer6.ai

## Abstract

As deep generative models have progressed, recent work has shown them to be capable of memorizing and reproducing training datapoints when deployed. These findings call into question the usability of generative models, especially in light of the legal and privacy risks brought about by memorization. To better understand this phenomenon, we propose the *manifold memorization hypothesis* (MMH), a geometric framework which leverages the manifold hypothesis into a clear language in which to reason about memorization. We propose to analyze memorization in terms of the relationship between the dimensionalities of $(i)$ the ground truth data manifold and $(ii)$ the manifold learned by the model. This framework provides a formal standard for "how memorized" a datapoint is and systematically categorizes memorized data into two types: memorization driven by overfitting and memorization driven by the underlying data distribution. By analyzing prior work in the context of the MMH, we explain and unify assorted observations in the literature. We empirically validate the MMH using synthetic data and image datasets up to the scale of Stable Diffusion, developing new tools for detecting and preventing generation of memorized samples in the process.

## 1 Introduction

Suppose $\{x_i\}_{i=1}^n$ is a dataset in $\mathbb{R}^d$ drawn independently from a ground truth probability distribution $p_*(x)$. A deep generative model (DGM) is a probability distribution $p_\theta(x)$ designed to capture $p_*(x)$ only from knowledge of $\{x_i\}_{i=1}^n$. DGMs, and most famously, diffusion models (DMs; Sohl-Dickstein et al., 2015; Ho et al., 2020), have led the "generative AI" boom with their ability to generate realistic images from text prompts (Karras et al., 2019; Rombach et al., 2022). DMs are thus likely to be deployed in an increasing number of public-facing or safety-critical applications. However, with sufficient model capacity, DGMs are known to memorize some of their training data. Memorization occurs at various degrees of specificity, including identities of brands, layouts of specific scenes, or exact copies of images (Webster et al., 2021; Somepalli et al., 2023a; Carlini et al., 2023).

Memorization is undesirable for myriad reasons. Simply put, the more a model reproduces its training data, the less useful it becomes. Memorization is a modelling failure under the DGM definition provided above; if the underlying ground truth $p_*(x)$ does not place positive probability mass on individual datapoints, then a $p_\theta(x)$ that memorizes any datapoint must be failing to generalize (Yoon et al., 2023). But memorization's risks go beyond mere utility. Training datasets may contain private information which, if memorized, might be exposed in downstream applications. Copyright law includes "substantial similarity" between generated and training data as a criterion in its definition of infringement, meaning that reproduced training samples can open up model builders or users to legal liability. For instance, the recent legal decision by Orrick (2023) hinged on this criterion.

The increasing dependence of society on generative models and resulting risks call for work to better understand memorization. Recent empirical work has identified mechanistic causes of memorization including but not limited to data complexity, duplication of training points, and highly specific labels (Somepalli et al., 2023b; Gu et al., 2023). We group these insights under the umbrella of "memorization phenomena", a catch-all term for the various interesting memorization-related observations we would like to understand better. Though useful in practice, these memorization

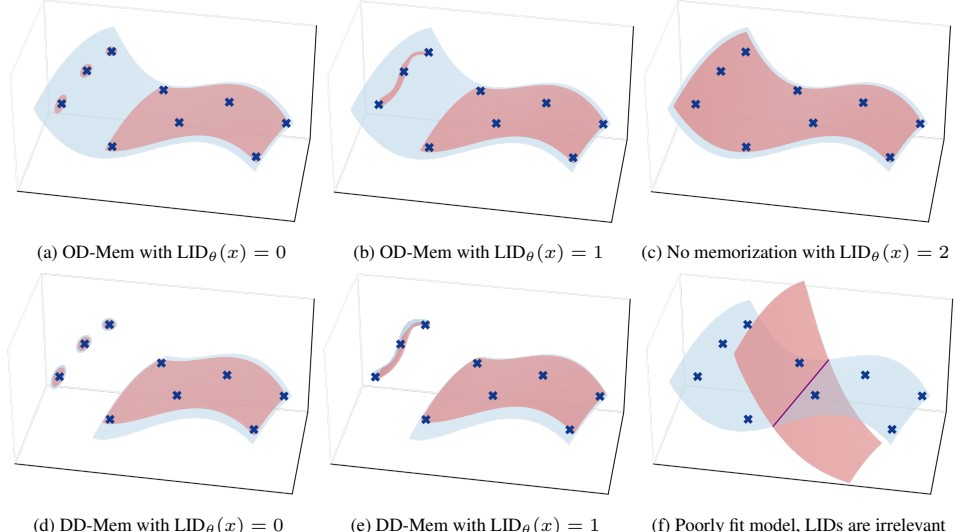

(a) OD-Mem with $\text{LID}_\theta(x) = 0$     (b) OD-Mem with $\text{LID}_\theta(x) = 1$     (c) No memorization with $\text{LID}_\theta(x) = 2$

(d) DD-Mem with $\text{LID}_\theta(x) = 0$     (e) DD-Mem with $\text{LID}_\theta(x) = 1$     (f) Poorly fit model, LIDs are irrelevant

Figure 1: An illustrative example of LID values for models with different quality of fit and degrees of memorization. In these plots, the ground truth manifold $\mathcal{M}_*$ is depicted in light blue, training samples $\{x_i\}_{i=1}^n \subset \mathcal{M}_*$ are depicted as crosses, and the model manifolds $\mathcal{M}_\theta$ are depicted in red. In (a) and (d), the model assigns 0-dimensional point masses around the three leftmost datapoints, indicating that it will reproduce them directly at test time; however in the former case this is caused by overfitting ($\text{LID}_\theta(x) < \text{LID}_*(x)$), while in the latter case it is caused by the ground truth data having small LID. The models in (b) and (e) are analoguous to (a) and (b), respectively, and still memorize, but with an extra degree of freedom in the form of a 1-dimensional submanifold containing the three points. Only the model in (c), which has learned a 2-dimensional manifold through its full support, has generalized well enough and has learned a manifold of high enough dimension to avoid both types of memorization. Finally, (f) shows a poorly fit model where LID and memorization are not meaningfully related.

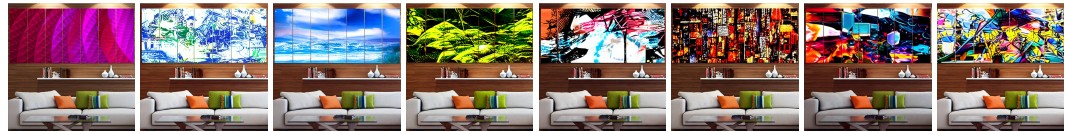

Figure 2: 8 images along a relatively low-dimensional manifold learned by Stable Diffusion v1.5. The first is a real image from LAION (flagged as memorized by Webster (2023)), and the remainder were generated by the model.

phenomena have yet to be unified and interpreted under a single theoretical framework. Meanwhile, formal treatments of memorization have led to isolated usecases such as detection (Meehan et al., 2020; Bhattacharjee et al., 2023) and prevention on a model level (Vyas et al., 2023), but have provided little explanatory power for memorization phenomena. In addition to providing theoretical insights, a unifying framework could yield more capabilities such as identifying whether a training image has been memorized, altering the sampling process to reduce memorization, and detecting memorized generations post hoc.

In this work, we introduce the *manifold memorization hypothesis (MMH)*, a geometric framework to explain memorization. In short, we propose that *memorization occurs at a point $x \in \mathbb{R}^d$ when the manifold learned by the generative model contains $x$ but has too small a dimensionality at $x$.* As we will see, this understudied perspective is a natural take on memorization that leads to practical insights and effectively explains memorization phenomena like those mentioned above. Although we mainly focus on DMs, the most notorious memorizers, our geometric framework applies to any DGM on a continuous data space $\mathbb{R}^d$; indeed, we empirically validate it on generative adversarial networks (GANs; Goodfellow et al., 2014; Karras et al., 2019) as well. Pidstrigach (2022) was the first to show that DMs are capable of learning low-dimensional structure in $\mathbb{R}^d$ and that this manifold learning capability is a driver of memorization; in this sense, our work extends this connection into a general framework, grounds it in empirical findings, and connects it to recent work on memorization.

This paper is organized according to the following contributions.

1. We advance the MMH in Section 2. After defining the key notions of the data manifold and local intrinsic dimension (LID), we describe how LIDs correspond to memorization.

2. We demonstrate the explanatory power of the MMH in Section 3 by grounding it in prior observations about the behaviour of models that memorize. As this section will show, memorization phenomena observed in past work can be predicted and explained by the MMH.

3. In Subsection 4.1, we empirically test the MMH, showing that it both accurately describes reality and is useful in practice. As predicted by the MMH, estimates of LID are strongly predictive of memorization at scales ranging from 2-dimensional synthetic data to Stable Diffusion (Rombach et al., 2022).

4. Finally, inspired by the MMH, in Subsection 4.2 we devise scalable approaches to avert memorization during sampling from Stable Diffusion and to identify tokens in the text conditioning that contribute to memorization.

## 2 Understanding Memorization through LID

**Preliminaries** Here we presume the manifold hypothesis: that data of interest lies on a manifold $\mathcal{M} \subset \mathbb{R}^d$ (Bengio et al., 2013). We take a generalized definition of manifold in which $\mathcal{M}$ is allowed to have different dimensionalities in different regions,[1] which is appropriate for realistic, heterogeneous data with varying degrees of structure and complexity. In particular, we assume that both our ground truth distribution $p_*(x)$ and our model $p_\theta(x)$ produce samples on manifolds, which we refer to as $\mathcal{M}_*$ and $\mathcal{M}_\theta$ respectively. We direct readers to Loaiza-Ganem et al. (2024) for a justification and formal mathematical treatment of both of these assumptions, which are especially valid when the data is high-dimensional and the models are high-performing ones such as DMs and GANs.

Our framework for understanding memorization revolves around the notion of a point's *local intrinsic dimension* (LID). Given a manifold $\mathcal{M}$ and a point $x \in \mathcal{M}$, we define the LID of $x$, $\text{LID}(x)$, with respect to $\mathcal{M}$ as the dimensionality of $\mathcal{M}$ at $x$. In this work, we will mainly consider the LIDs of points $x \in \mathbb{R}^d$ with respect to two specific manifolds: $\mathcal{M}_*$ and $\mathcal{M}_\theta$. We will refer to these quantities as $\text{LID}_*(x)$ and $\text{LID}_\theta(x)$, respectively.

**Intuition and the Manifold Hypothesis** Before discussing our framework, we review some intuition relating the manifold hypothesis to practical datasets. Manifold structure $\mathcal{M} \subset \mathbb{R}^d$ arises from sets of constraints. These can range from very simple, like a set of linear constraints ($\mathcal{M} = \{x \mid Ax = b\}$), to highly complex ($\mathcal{M} = \{x \mid x \text{ is an image of a face}\}$). Locally at a point $x \in \mathcal{M}$, each constraint determines a direction one cannot move without leaving the manifold and violating the structure of the dataset.[2] Hence, a region governed by $\ell$ independent and active constraints will have dimensionality $\text{LID}(x) = d - \ell$. The value of $\text{LID}(x)$ can be intuited as the number of degrees of freedom – valid independent directions of movement in which the characteristics of the dataset are preserved. Another connection is to complexity. For example, estimates of LID from algorithms like FLIPD (Kamkari et al., 2024b) or the normal bundle (NB) method of Stanczuk et al. (2024) (which we use in our experiments; see Appendix B for details) have been shown to correspond closely with the complexity of an image; it is reasonable to expect that images with more complex features can endure more changes (such as morphing, moving, or changing the colours of different parts of the image) without losing coherence. The notions of constraints, degrees of freedom, and complexity along with their relationship to LID will help us understand its connection to memorization in later sections.

**A Geometric Framework for Understanding Memorization** In this section we formulate a framework for understanding memorization based on comparisons between $\text{LID}_\theta(x)$ and $\text{LID}_*(x)$. As a motivating example, consider Figure 1, which depicts six possible models $p_\theta(x)$ trained on datasets $\{x_i\}_{i=1}^n$ that each lie on a ground truth manifold $\mathcal{M}_*$. In the first scenario, Figure 1a, the

---

[1]Most authors define a manifold to have a constant dimension over the entire set. Under this common definition, our assumption is referred to as the union of manifolds hypothesis (Brown et al., 2023). We use a more general definition of manifold for brevity.

[2]This statement is captured formally by the regular level set theorem of differential geometry, and manifolds can be modelled as such (Lee, 2012; Ross et al., 2023).

model $p_\theta(x)$ has precisely memorized some of the training data. This is a well-understood mode of memorization; training datapoints are exactly reproduced. To achieve this, the model has learned a 0-dimensional manifold around these datapoints. To our knowledge, Pidstrigach (2022) was the first to point out that a model capable of learning 0-dimensional manifolds can memorize the training data. From this example, we infer that $x$ can be perfectly reproduced when $\text{LID}_\theta(x) = 0$. This indicates suboptimality in the model at the datapoints shown, for which $\text{LID}_*(x) = 2$.

However, memorization can be more complex than simply reproducing a datapoint. For example, Somepalli et al. (2023a) identify instances where layouts, styles, or foreground or background objects in training images are copied without copying the entire image, a phenomenon they refer to as *reconstructive memory*. Webster (2023) surfaces more instances of the same phenomenon and refers to them as *template verbatims*. See Figure 2 for an example. In the region of these points $x \in \mathcal{M}_\theta$, the model is able to generate images with degrees of freedom in some attributes (e.g., colour or texture), but is too constrained in other attributes (e.g., layout, style, or content). Geometrically, $\mathcal{M}_\theta$ is too constrained compared to the idealized ground truth manifold $\mathcal{M}_*$; i.e., $\text{LID}_\theta(x) < \text{LID}_*(x)$. We depict this situation in Figure 1b, wherein the model has erroneously assigned $\text{LID}_\theta(x) = 1$ for some of the training datapoints.

**Two Types of Memorization**  We expect two types of memorization to be of interest. An academic interested in designing DGMs that learn the ground truth distribution correctly will chiefly be interested in avoiding the memorization scenario $\text{LID}_\theta(x) < \text{LID}_*(x)$. We refer to this first scenario as *overfitting-driven memorization* (OD-Mem). This situation represents a modelling failure in that $p_\theta(x)$ is not generalizing correctly to $p_*(x)$, and is illustrated in Figure 1a and Figure 1b.

However, an industry practitioner deploying a consumer-facing model might be more interested in hypothetical values of $\text{LID}_\theta$ *per se*, irrespective of the values of $\text{LID}_*$. For any points $x \in \mathcal{M}_*$ containing trademarked or private information, low values of $\text{LID}_\theta(x)$ will be of concern even if $\text{LID}_\theta(x) = \text{LID}_*(x)$, as this information is likely to be revealed in samples generated from this region. A practitioner would rightly refer to this situation as memorization despite the model generalizing correctly. We refer to this second scenario as *data-driven memorization* (DD-Mem), and illustrate it in Figure 1d and Figure 1e. This certainly happens in practice; for example, conditioning on the title of a specific artwork (e.g. "*The Great Wave off Kanagawa*" by Katsushika Hokusai (Somepalli et al., 2023a)) is a very strong constraint, leaving few degrees of freedom in the ground truth manifold $\mathcal{M}_*$, but reproducing specific artworks may be undesirable in a production model. Unlike OD-Mem, DD-Mem is not overfitting in the classical sense, and a notable consequence is that it cannot be detected by comparing training and test likelihoods. We refer to the conceptualization of how LIDs relate to memorization through OD-Mem and DD-Mem as the *manifold memorization hypothesis*.

No memorization is present in Figure 1c, in which the model manifold $\mathcal{M}_\theta$ matches the ground truth manifold $\mathcal{M}_*$ and both have sufficiently high LID. We highlight that the MMH assumes high-performing models whose manifold $\mathcal{M}_\theta$ is roughly aligned with the data manifold $\mathcal{M}_*$; when this is not the case, as in Figure 1f, $\text{LID}_\theta$ and its relationship to $\text{LID}_*$ become irrelevant to memorization.

**Why is the MMH Useful?**  The MMH is a hypothesis about how memorization occurs in practice for high-dimensional data. Its utility is best framed in contrast to past treatments of memorization. First, while past theoretical frameworks for memorization have focused on probability mass, our geometric perspective leads to more practical tools. For example, Bhattacharjee et al. (2023) propose a purely probabilistic definition of memorization that can be detected only with access to the training dataset and the ability to generate large numbers of samples, which are intractable requirements at the scale of LAION-2B (Schuhmann et al., 2022) and Stable Diffusion. In contrast, the MMH suggests that memorization can be detected through $\text{LID}_\theta(x)$, for which tractable estimators exist at scale. We explore these estimators in Section 4.

Second, the MMH explains and quantifies the phenomenon depicted in Figure 2: reconstructive memorization. While it has been studied in the past (Somepalli et al., 2023a; Webster, 2023; Wen et al., 2023), it has been resistant to theoretical explanation in part because past work has defined memorization based on distance to the memorized training point (see Appendix A for more discussion on definitions). It is clear from Figure 2 that distance cannot capture reconstructive memorization; the training datapoint on the left is far in pixel space from the Stable Diffusion-generated samples to its right. Our framework overcomes this challenge by interpreting memorization in relation to the model and data manifolds without reference to distances or any specific training datapoint.

Third, the MMH distinguishes between OD-Mem and DD-Mem, while past analyses have not. Bhattacharjee et al. (2023) would allow for OD-Mem but not DD-Mem under their definition of memorization, while empirical work tends to ignore the effect of $p_*(x)$ on $p_\theta(x)$, thus subsuming both OD-Mem and DD-Mem in spirit if not formally (Carlini et al., 2023; Yoon et al., 2023; Gu et al., 2023). For further details, please see Appendix A, where we formally develop the relationship between the MMH and definitions of memorization in related work.

Defining and distinguishing between OD-Mem and DD-Mem suggests immediate solutions to each. OD-Mem is overfitting and can be addressed accordingly, such as by collecting more data or improving a model's inductive biases. On the other hand, DD-Mem indicates that the training distribution $p_*(x)$ does not actually match the desired distribution at inference time, and hence is a misalignment of data and objectives. It can be addressed by changing $p_*(x)$ itself, such as by altering the data collection, cleaning, and augmentation procedures. We explore this point further in Section 3. Unlike OD-Mem, DD-Mem cannot be addressed by improving the model to generalize better. Both OD-Mem and DD-Mem can also in principle be addressed by augmenting $p_\theta(x)$ to generate higher-LID samples. In Section 4, we propose solutions to alter the data-generating process with precisely this goal.

## 3    Explaining Memorization Phenomena

In this section, we demonstrate the explanatory power of the MMH by showing how it explains memorization phenomena in related work. In the process, this section demonstrates two advantages of our geometric framework. First, it provides a unifying perspective on seemingly disparate observations throughout the literature (nevertheless, this is not meant as a related work section — for that see Section 5). Second, the MMH links memorization to the rich theoretical toolboxes of measure theory and geometry, which we use in this section to establish formal connections to past work. Propositions, theorems, and proofs in this section are presented informally for clarity. For full theorem statements and proofs, please see Appendix E.

**Duplicated Data and LID**    It has been broadly observed that memorization occurs when training points are duplicated (Nichol et al., 2022; Carlini et al., 2022; Somepalli et al., 2023a). In Proposition 3.1, we show that duplicated datapoints lead to DD-Mem; duplicated points $x_0$ indicate $\text{LID}_*(x_0) = 0$, so even a correctly fitted model will have $\text{LID}_\theta(x_0) = 0$ (as in Figure 1d).

**Proposition 3.1** (Informal). *Let $\{x_i\}_{i=1}^n$ be a training dataset drawn independently from $p_*(x)$. Under some regularity conditions, the following hold:*

1. *If duplicates occur in $\{x_i\}_{i=1}^n$ with positive probability, then they occur at a point $x_0$ such that $\text{LID}_*(x_0) = 0$.*

2. *If $\text{LID}_*(x_0) = 0$ and $n$ is sufficiently large, then duplication will occur in $\{x_i\}_{i=1}^n$ with near-certainty.*

*Proof.* See Appendix E for the formal statement of the theorem and proof. To understand both conditions, it suffices to note first that duplicate samples are intuitively equivalent to $p_*(x)$ assigning positive probability to a point. Under mild regularity conditions on the nature of the $p_*(x)$ and $\mathcal{M}_*$, positive probability at a point is equivalent to a 0-dimensional manifold at that point.  □

From this result, we gather that improving model generalization is not the solution to duplication. Instead, one may need to add inductive biases that prevent $p_\theta(x)$ from learning 0-dimensional points. Of course, the more straightforward path is to change the data distribution $p_*(x)$ by de-duplicating the training dataset. We carry the same intuition forward to "near-duplicated content", where similar but non-identical points occur together in the dataset, in which case $\text{LID}_*$ would be low but nonzero in the region of the near-duplicated content (as in Figure 1e).

**Conditioning and LID**    Somepalli et al. (2023b) and Yoon et al. (2023) observe that conditioning on highly specific prompts $c$ encourages the generation of memorized samples. Here, we point out that conditioning decreases LID, making models more likely to generate memorized samples.

**Proposition 3.2** (Informal). *Let $x_0 \in \mathcal{M}_*$, and let us denote by $\text{LID}_*(x_0 \mid c)$ the LID of $x_0$ with respect to the support of the conditional distribution $p_*(x \mid c)$. We then have*

$$\text{LID}_*(x_0 \mid c) \leq \text{LID}_*(x_0). \tag{1}$$

*Proof.* See Appendix E for the formal statement of the theorem and proof. Intuitively, conditioning can be interpreted as adding additional constraints to $\mathcal{M}_*$, which cannot increase its dimension. $\quad\square$

Conditioning on highly specific $c$ can be linked to both DD-Mem and OD-Mem. Introducing strong constraints greatly decreases $\text{LID}_*$, leading to DD-Mem. However, if a relatively low number of training examples satisfy $c$, the model could overfit, leading to OD-Mem as well.

**Complexity and LID**   For images, Somepalli et al. (2023b) also highlight low complexity as a factor causing memorization. Using the understanding that LID corresponds to complexity as discussed in Section 2, we infer that low-complexity datapoints $x \in \mathcal{M}_*$ have low $\text{LID}_*(x)$. This fact suggests that, like with duplication, memorization of low-complexity datapoints is an example of DD-Mem.

**The Classifier-Free Guidance Norm and LID** Classifier-free guidance (CFG) is a way to improve the quality of conditional generation in DMs. Whereas standard conditional generation employs the score function $s_\theta(x; t, c)$, which refers to a neural estimate at time $t$ of the conditional score, CFG increases the strength of conditioning by using the following modified score:

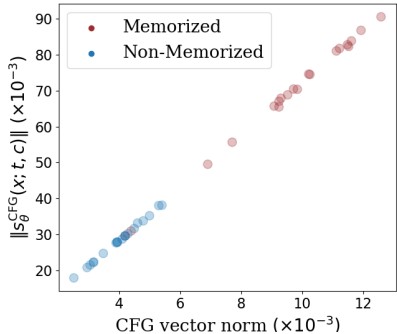

$$\underbrace{s_\theta^{\text{CFG}}(x; t, c)}_{\text{CFG-adjusted score}} = s_\theta(x; t, \emptyset) + \lambda(\underbrace{s_\theta(x; t, c) - s_\theta(x; t, \emptyset)}_{\text{CFG vector}}), \quad (2)$$

where $\lambda$ is a hyperparameter for "guidance strength" and $s_\theta(x; t, \emptyset)$ refers to conditioning on the empty string (here we formulate DMs using stochastic differential equations (Song et al., 2021)).

Figure 3: CFG-adjusted scores vs CFG vectors for Stable Diffusion with $\lambda = 7.5$ and $t = 0.02$ on 20 memorized and 20 non-memorized images from LAION.

Wen et al. (2023) identify that specific conditioning inputs $c$ lead to memorized samples when the CFG vector has a large magnitude. We explain this observation using the MMH as follows. First, we observe that a large CFG magnitude will generally result in a large magnitude of the CFG-adjusted score $s_\theta^{\text{CFG}}(x; t, c)$. We demonstrate this empirically in Figure 3. Furthermore, it is understood in the literature that a large $\|s_\theta^{\text{CFG}}(x; t, c)\|$, and its explosion as $t \to 0$, is common for high-dimensional data (Vahdat et al., 2021) and is necessary to generate samples from low-dimensional manifolds (Pidstrigach, 2022; Lu et al., 2023). It has been empirically observed that this explosion occurs faster as the dimensionality gap increases between the data manifold and the ambient data space (Loaiza-Ganem et al., 2024), which is one reason that generative modelling on lower-dimensional latent space tends to improves performance (Loaiza-Ganem et al., 2022). The largest $\|s_\theta^{\text{CFG}}(x; t, c)\|$ values should thus generate points with the largest dimensionality difference from $\mathbb{R}^d$; i.e., points $x$ with the smallest $\text{LID}_\theta(x \mid c)$. [3] Hence we infer that reducing the CFG-adjusted score norm – or equivalently the CFG vector norm – should increase $\text{LID}_\theta(x \mid c)$ and lessen memorization, a fact confirmed empirically by Wen et al. (2023). Since this phenomenon corresponds to any $x$ with small $\text{LID}_\theta(x \mid c)$, it can indicate both OD-Mem and DD-Mem under the MMH.

## 4   EXPERIMENTS

### 4.1   VERIFYING THE MANIFOLD MEMORIZATION HYPOTHESIS

In this section, we empirically verify the geometric framework which underpins the MMH. We analyze both $\text{LID}_*$ and $\text{LID}_\theta$ to study DD-Mem and OD-Mem. Several algorithms exist to estimate $\text{LID}_\theta(x)$ for diffusion models, including the normal bundle (NB) method (Stanczuk et al., 2024), and more recently FLIPD (Kamkari et al., 2024b). For GANs, we approximate $\text{LID}_\theta(x)$ of generated data by computing the rank of the Jacobian of the generator. Additionally, we use LPCA (Fukunaga & Olsen, 1971) to estimate $\text{LID}_*$ where applicable; see Appendix B for details on these methods and Appendix C for their hyperparameter configurations. In general $\text{LID}_\theta$ and $\text{LID}_*$ are unknown quantities that are approximated with the aforementioned estimators, throughout this section we write their respective estimates as $\widehat{\text{LID}}_\theta$ and $\widehat{\text{LID}}_*$.

---

[3]Here we understand $\text{LID}_\theta(x \mid c)$ as the LID with respect to the support of the conditional distribution being sampled from by $s_\theta^{\text{CFG}}(x; t, c)$.

**Diffusion Model on a von Mises Mixture** In an illustrative experiment, we study a mixture of a von Mises distribution, which sits on a 1-dimensional circle, and a 0-dimensional point mass at the origin in 2-dimensional ambient space, as depicted in Figure 4; every point $x \in \mathcal{M}_*$ has either $\text{LID}_*(x) = 0$ or $\text{LID}_*(x) = 1$. From this distribution we sample 100 training points, and by chance a single point $x_0$ sits isolated in a low-density region of the circle. Next, we train a DM on this data. In Figure 4 we depict 100 generated samples, colour-coded by their LID estimates, as estimated by FLIPD. Here, we see OD-Mem and DD-Mem in action: the model overfits at $x_0$, producing near-exact copies, with $0 \approx \widehat{\text{LID}}_\theta(x_0) < \text{LID}_*(x_0) = 1$ (OD-Mem). The model also faithfully produces copies of the circle's center, but this is caused by low ground truth LIDs (DD-Mem), not modelling error.

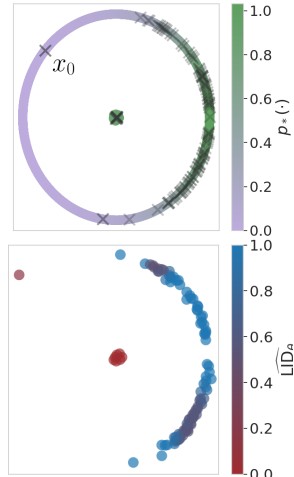

Figure 4: Training a diffusion model on a von Mises mixture. (Top) Ground truth manifold and the associated distribution. (Bottom) Model-generated samples with their LID estimates.

**CIFAR10 Memorization** We analyze the higher-dimensional CIFAR10 dataset (Krizhevsky & Hinton, 2009) and use two pretrained generative models: iDDPM (Nichol & Dhariwal, 2021) and StyleGAN2-ADA (Karras et al., 2020). We generate 50,000 images from each model, and for each, we identify the most similar training image according to two distance metrics, $(i)$ SSCD distance (Pizzi et al., 2022) and $(ii)$ calibrated $\ell_2$ distance (Carlini et al., 2023). By thresholding on these metrics, we arrive at a small subset of potentially memorized examples, which we manually label as either exactly memorized, reconstructively memorized (Somepalli et al., 2023a), or not memorized. All other images are not labelled and have a low chance of being memorized. Further details and all images deemed memorized are reported in Appendix F. The first two panels in Figure 5a show our labels distinguish different types of memorization as we display the generated images vs. the closest SSCD match in the training dataset.

Next, we estimate $\text{LID}_\theta$ for each iDDPM and StyleGAN2-ADA sample. For iDDPM, we use the NB estimator. Figure 5b and Figure 5c show that $\widehat{\text{LID}}_\theta$ is generally smaller for memorized images compared to non-memorized ones. As shown in Figure 5d, $\widehat{\text{LID}}_*$ is considerably lower for exact memorization cases within the training dataset, suggesting that exact memorization for both models corresponds to DD-Mem. We also observe that in Figure 5b, reconstructively memorized samples exhibit lower values of $\widehat{\text{LID}}_\theta$ as compared to not memorized samples, despite the corresponding training data having comparable $\widehat{\text{LID}}_*$ (Figure 5d): the $\text{LID}_\theta$ estimates enable us to still classify these samples as memorized, showing a clear example of detecting OD-Mem.

We have shown that LID estimates are effective at detecting both OD-Mem and DD-Mem, supporting the MMH hypothesis. However, while simpler images tend to be memorized more frequently, they are not always memorized (see Figure 5a, right panel), leading to some overlap in estimated $\text{LID}_\theta$ between memorized and not memorized samples in Figure 5b and Figure 5c. This overlap occurs

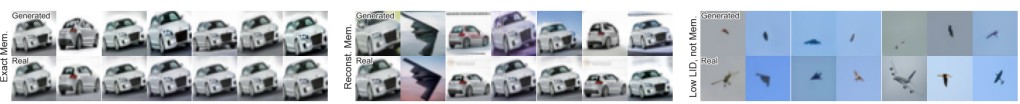

(a) (Left) Exact and (Middle) reconstructively memorized samples (Top) with their matched CIFAR10 datapoints (Bottom). (Right) Non-memorized samples with low $\widehat{\text{LID}}_\theta$, showing $\widehat{\text{LID}}_\theta$ can be partially confounded by complexity.

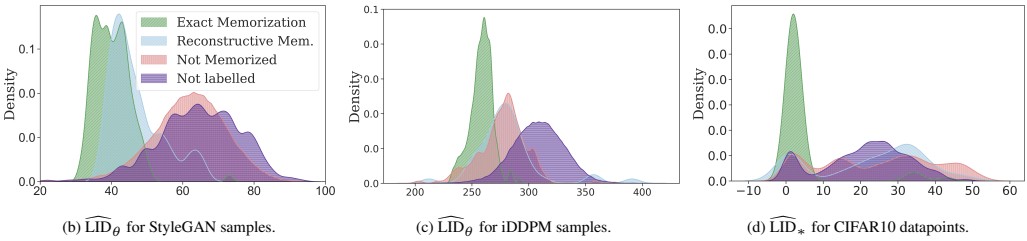

(b) $\widehat{\text{LID}}_\theta$ for StyleGAN samples.  (c) $\widehat{\text{LID}}_\theta$ for iDDPM samples.  (d) $\widehat{\text{LID}}_*$ for CIFAR10 datapoints.

Figure 5: Visualizing OD-Mem and DD-Mem on StyleGAN2-ADA and iDDPM trained on CIFAR10.

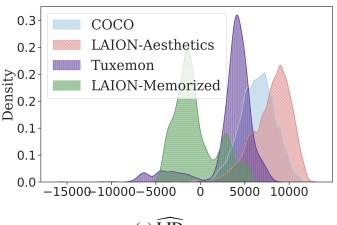 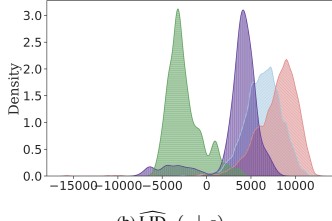 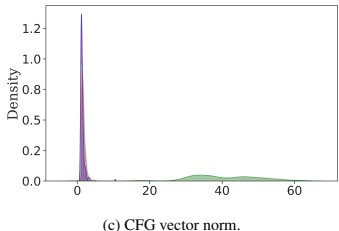

(a) $\widehat{\mathrm{LID}}_\theta$.        (b) $\widehat{\mathrm{LID}}_\theta(\cdot \mid c)$.        (c) CFG vector norm.

Figure 6: Density histograms for each memorization metric across different datasets.

because image complexity serves as a confounding factor: images with simple backgrounds and textures may be assigned low $\mathrm{LID}_\theta$ values, not due to memorization, but simply because of their inherent simplicity. We discuss this issue further, along with a partial solution, in Appendix C.2.

**Stable Diffusion on Large-Scale Image Datasets** Here, we set $p_\theta(x)$ to Stable Diffusion v1.5 (Rombach et al., 2022). Taking inspiration from the benchmark of Wen et al. (2023), we retrieve memorized LAION (Schuhmann et al., 2022) training images identified by Webster (2023). We focus on the 86 memorized images categorized as "matching verbatim", noting that the other categories of Webster (2023) consist of large numbers of captions that generate samples matching a small set of training images. For non-memorized images, we use a mix of 2000 images sampled from LAION Aesthetics 6.5+, 2000 sampled from COCO (Lin et al., 2014), and all 251 images from the Tuxemon dataset (Tuxemon Project, 2024; Hugging Face, 2024).

To our knowledge, no estimator of $\mathrm{LID}_*$ scales to images at the size of Stable Diffusion; we thus omit these from our analysis. FLIPD is the only $\mathrm{LID}_\theta$ estimator that remains tractable at this scale, so we use it for this analysis. Note that Stable Diffusion provides two model distributions: the unconditional distribution $p_\theta(x)$ and the conditional distribution $p_\theta(x \mid c)$, where $c$ is the image's caption. Hence, we compute both $\widehat{\mathrm{LID}}_\theta$ and $\widehat{\mathrm{LID}}_\theta(\cdot \mid c)$ for each of the aforementioned images. Additionally, we compute the norm of the CFG vector, which was proposed as a memorization detection method by Wen et al. (2023) and which we argued varies inversely to $\mathrm{LID}_\theta$ in Section 3. Our experiments thus cover three proxies for local intrinsic dimension: $\widehat{\mathrm{LID}}_\theta$, $\widehat{\mathrm{LID}}_\theta(\cdot \mid c)$, and the CFG vector norm (see Appendix C for details). The density histograms of all these values are depicted in Figure 6.[4]

We see that all proxies for $\mathrm{LID}_\theta$ assign relatively small LID values to memorized images, further validating the MMH. Due to the unavailability of $\mathrm{LID}_*$ estimates, it is hard to distinguish between DD-Mem and OD-Mem here. In Figure 6, low conditional or unconditional LID as well as high CFG vector norms are all signals of memorization, strengthening our argument in Section 3. While the CFG vector norm seemingly provides the strongest signal, the unconditional LID detects memorization well despite lacking caption information. Detecting memorized training images without the corresponding captions is a novel capability, and notably cannot be done with the CFG vector norm technique.

## 4.2 MITIGATING MEMORIZATION BY CONTROLLING $\mathrm{LID}_\theta$

In this section we study the problem of sample-time mitigation through the lens of the MMH. Somepalli et al. (2023b) establish text-conditioning as a crucial driver of memorization in Stable Diffusion, where specific tokens in the prompt often cause the model to generate replicas of training images. Wen et al. (2023) introduce a differentiable metric, which we denote as $\mathcal{A}^{\mathrm{CFG}}(c)$ (formally defined in Appendix D), which is based on the accumulated CFG vector norm while sampling an image. Wen et al. (2023) observe that this metric shows a sharp increase when the prompt $c$ leads to the generation of memorized images. Since $\mathcal{A}^{\mathrm{CFG}}(c)$ is differentiable with respect to $c$, Wen et al. (2023) backpropagate through this metric and find the tokens with the largest gradient magnitude, essentially providing token attributions for memorization.

Here we make two contributions. First, we propose two additional metrics, $\mathcal{A}^{s_\theta^{\mathrm{CFG}}}(c)$ and $\mathcal{A}^{\mathrm{FLIPD}}(c)$, which are modifications of $\mathcal{A}^{\mathrm{CFG}}(c)$ to use $\|s_\theta^{\mathrm{CFG}}(x; t, c)\|$ or FLIPD, respectively, instead of the

---

[4]The LID estimates provided by FLIPD are sometimes negative in value; Kamkari et al. (2024b) justify this as an artifact of estimating the LID using a UNet. Despite underestimating LID in absolute terms, Kamkari et al. (2024b) confirm that FLIPD ranks $\mathrm{LID}_\theta$ estimates correctly, which is sufficient for the purpose of distinguishing memorized from non-memorized examples.

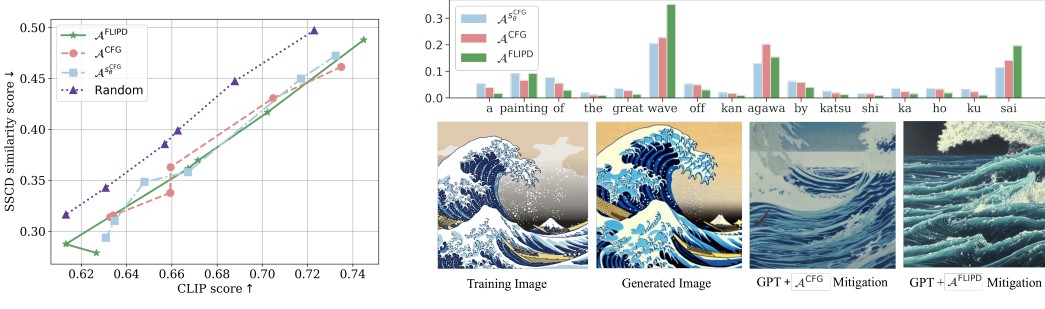

(a) Analysis of the mitigation approach.    (b) Comparing (normalized) token attributions for a memorized prompt using three methods.

Figure 7: Using token attributions to detect drivers of memorization and to mitigate it at sample time.

norm of the CFG vector. We define these metrics fully in Appendix D due to space limitations. Since both of these new metrics are also differentiable with respect to $c$, $\mathcal{A}^{\mathrm{CFG}}(c)$ can be trivially replaced by either of them in the method of Wen et al. (2023). Second, we propose an automated way to use the token attributions from this method into a sample-time mitigation scheme. We start by normalizing the attributions across the tokens, and sample $k$ tokens based on a categorical distribution parameterized by these normalized attributions. We then use GPT-4 (OpenAI, 2023) to rephrase the caption, keeping it semantically similar but perturbing the selected $k$ tokens that are highly contributing to the memorization metric (see Appendix D.4 for details).

The bottom panel in Figure 7b shows four images: a training image corresponding to the prompt "*The Great Wave off Kanagawa by Katsushika Hokusai*", a generated image using the same prompt showing clear memorization, a generated image obtained with our mitigation scheme with $\mathcal{A}^{\mathrm{CFG}}(c)$, and another generated image using $\mathcal{A}^{\mathrm{FLIPD}}(c)$ instead. Qualitatively, using FLIPD or the norm of the CFG vector perform on par with each other. The top panel of Figure 7b shows the token attributions obtained from $\mathcal{A}^{\mathrm{FLIPD}}(c)$ are sensible. See Appendix D.5 for additional results.

We present quantitative comparisons in Figure 7a by analyzing the average CLIP score (higher is better) (Radford et al., 2021) and SSCD similarity (lower is better) over matching prompts, varying $k \in \{1, 2, 3, 4, 6, 8\}$, with 5 repetitions for each prompt. As $k$ increases, both SSCD and CLIP scores decrease across methods. We also include an ablation where the modified tokens are selected uniformly at random, ignoring attributions. All attribution-based methods achieve lower similarity while maintaining a relatively higher CLIP score than uniform token selection.

Overall, the results in Figure 7 provide further evidence supporting the MMH, both by showing that encouraging samples to have higher LID can help prevent memorization, and by further confirming the relationship between the CFG vector norm, the CFG-adjusted score norm, and LID established in Section 3. We hypothesize that our results can likely be improved by more efficiently guiding generated samples towards regions of high LID, but highlight that doing so is not trivial. For example, in Appendix D.3 we find that optimizing $c$ for large values of $\mathcal{A}^{\mathrm{FLIPD}}(c)$ (inspired by the inference-time mitigation method of (Wen et al., 2023)) during sampling can fail by producing samples with chaotic textures that have artificially high $\mathrm{LID}_\theta$.

## 5 RELATED WORK

**Detecting and Preventing Memorization for Image Models**   The task of surfacing memorized samples is well-studied. Consensus in the literature is that $\ell_2$ distance to the nearest training sample in pixel space is a poor detector of memorized samples (Carlini et al., 2023), but that recalibrating the $\ell_2$ distance according to the local concentration of the dataset works better for smaller datasets (Yoon et al., 2023; Stein et al., 2023), and that using retrieval techniques such as distance in SSCD feature space (Pizzi et al., 2022) works better still, especially for more complex, higher-resolution images (Somepalli et al., 2023a). However, all of these retrieval techniques are too expensive to be used to withhold samples from a live model. To more efficiently prevent memorized samples from being generated, past and concurrent works have altered the sampling procedure, training procedure, or the model itself (Wen et al., 2023; Daras et al., 2024; Chen et al., 2024; Hintersdorf et al., 2024).

**Explaining Memorization** There is an active community effort attempting to explain why and how memorization occurs in DGMs. Early studies focused on GANs, and have taken both theoretical (Nagarajan et al., 2018) and empirical (Bai et al., 2021) perspectives. However, GANs are thought to be less prone to memorization than DMs (Akbar et al., 2025), except on small datasets (Feng et al., 2021). Several works on DMs (Pidstrigach, 2022; Yi et al., 2023; Gu et al., 2023; Li et al., 2024) have pointed out that, given sufficient capacity, DMs at optimality are capable of learning the empirical training distribution, which is complete memorization. Others have focused on generalization, showing that DMs are capable of generalizing well in theory (Li et al., 2023), have inductive biases towards generating photorealistic images (Kadkhodaie et al., 2024), and will generalize when their capacity is insufficient to memorize (Yoon et al., 2023).

**DGM-Based LID Estimation** As opposed to statistical LID estimators (e.g., Levina & Bickel (2004)), which are constructed to estimate the dimension of $\mathcal{M}_*$, DGM-based ones estimate the dimensionality of $\mathcal{M}_\theta$, the manifold learned by a DGM. These types of estimators are available for many types of DGMs, and in addition to being useful for memorization, have found utility in out-of-distribution detection (Kamkari et al., 2024a). In the literature, LID estimators for normalizing flows (Dinh et al., 2014) have been proposed using the singular values of their Jacobians (Horvat & Pfister, 2022; Kamkari et al., 2024a) or their density estimates (Tempczyk et al., 2022). In Section 4 we applied the singular value method to obtain LID estimates for GANs. Dai & Wipf (2019) and Zheng et al. (2022) proposed estimators for VAEs (Kingma & Welling, 2014; Rezende et al., 2014) using the structure of their posterior distribution. Several authors have proposed estimators for DMs as well (Stanczuk et al., 2024; Kamkari et al., 2024b; Horvat & Pfister, 2024); we focus on those of Stanczuk et al. (2024) and Kamkari et al. (2024b) because they work with off-the-shelf DMs and do not require modifying the training procedure.

## 6 Conclusions, Limitations, and Future Work

Throughout this work, we have drawn connections between the geometry of a DGM and its propensity to memorize through the MMH. First, we showed that the notion of LID provides a systematic way of understanding different types of memorization. Second, we explained how memorization phenomena described by prior work can be understood from the perspective of LID. Third, we verified the MMH empirically across scales of data and classes of models. Fourth, we showed that controlling $\text{LID}_\theta$ is a promising way to mitigate memorization. We offered several connections, including the insight that some instances of memorization in DMs are due to the DM's inability to generalize (OD-Mem), whereas others are due to low-LID ground truth (DD-Mem).

Despite having shown that the MMH is a principled avenue to detect and alleviate memorization, our current approaches can be improved: estimates of $\text{LID}_\theta$ have some overlap between memorized and non-memorized samples and our sample-time scheme for mitigating memorization using $\mathcal{A}^{\text{FLIPD}}(c)$ performs on par with, but does not outperform, its more ad-hoc variant using $\mathcal{A}^{\text{CFG}}(c)$. We expect future work to find even better ways of leveraging the MMH and LID towards these goals; e.g. by improving LID estimation or more efficiently controlling LID during sampling. Finally, although the manifold hypothesis does not apply directly to discrete data such as language, some intuitions described in this work carry over, and generalizations or parallels to the concepts here may offer insights for the language-modelling space.

**Reproducibility Statement** To ensure the reproducibility of our experiments, we provide two codebase links. The first codebase, accessible at `github.com/layer6ai-labs/dgm_geometry`, contains our small-scale synthetic experiments and our CIFAR10 experiments. The second, accessible at `github.com/layer6ai-labs/diffusion_memorization/`, extends the work of Wen et al. (2023) to use the MMH to detect and mitigate memorization. Comprehensive details of our experimental setup are provided across Section 4, Appendix C, and Appendix D. All datasets used in our experiments are freely available from the referenced sources and are utilized in compliance with their respective licenses.

**Ethics Statement** We do not foresee any ethical concerns with this research. The overarching topic, memorization in generative models, is widely studied to better understand safety concerns associated with using and deploying such models. Our goal is to theoretically explain and to empirically detect and alleviate this phenomenon; we do not promote the use of these models for harmful practices.

ACKNOWLEDGMENTS

We thank Kin Kwan Leung for his valuable help in proving Lemma E.4.

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

## A CONTEXTUALIZING THE MMH WITHIN DEFINITIONS OF MEMORIZATION

**An Overview of Definitions**   The MMH describes the mechanism through which memorization occurs. How does this mechanism fit into prior definitions of memorization from the literature? Formal definitions of memorization generally follow the same template: a point $x_0$ is memorized when the model's probability measure $P_\theta$ places too much mass within some distance of $x_0$. Some of these definitions define memorization globally on the level of an entire model (Meehan et al., 2020; Yoon et al., 2023; Gu et al., 2023), while others define memorization locally for individual datapoints (Carlini et al., 2023; Bhattacharjee et al., 2023). The identical definitions of Yoon et al. (2023) and Gu et al. (2023) consider a point to be memorized based purely on a distance threshold; in practice, however, distances alone have been unsuccessful at consistently surfacing what would be perceived by humans as memorized (Somepalli et al., 2023a; Stein et al., 2023). We postulate this is also due to manifold structure; semantically memorized images such as Figure 2 will sit on the same manifold, but may not necessarily be close to each other as measured by distance, even when taken in the latent space of an encoder. Meanwhile, Carlini et al. (2023) take a privacy perspective; their definition considers images memorized if they can be extracted from a model by any means, not just generated by the model. In this work we take the perspective that memorized samples are most likely to be problematic when they are *generated* by a production model, which are often treated as a black-box, so we focus on generation rather than extraction.

**Links Between Formal Memorization and the MMH**   For the reasons above, we use here the definition of memorization by Bhattacharjee et al. (2023), who define a point $x_0$ as memorized by comparing $P_\theta$ to the ground truth $P_*$ in a neighbourhood of $x_0$. We present their definition here:

**Definition A.1.** Let $P_*$ and $P_\theta$ be the ground truth and model probability measures, respectively. Let $\lambda > 1$ and $0 < \gamma < 1$. A point $x \in \mathbb{R}^d$ is a $(\lambda, \gamma)$-copy of a training datapoint $x_0$ if there exists a radius $r > 0$ such that the $d$-dimensional ball $B_r^d(x_0)$ of radius $r$ centred at $x_0$ satisfies $(i)$ $x \in B_r^d(x_0)$, $(ii)$ $P_\theta(B_r^d(x_0)) \geq \lambda P_*(B_r^d(x_0))$, and $(iii)$ $P_*(B_r^d(x_0)) \leq \gamma$.

The first and third conditions imply that $x$ is sufficiently close to $x_0$ relative to the amount of probability mass in the region $(P_*(B_r(x_0)))$, while the second condition implies that the model $P_\theta$ places much more mass in the region compared to the ground truth $P_*$. A natural question about the MMH is whether points satisfying it are also formally memorized by the above definition. The answer is in the negative for DD-Mem (Proposition A.2) and the affirmative for OD-Mem (Theorem A.3).

**Proposition A.2.** *There exist models $p_\theta(x)$ that exhibit DD-Mem at $x_0 \in \mathbb{R}^d$, but do not generate $(\lambda, \gamma)$-copies of $x_0$.*

*Proof.* Choose any ground truth distribution $P_*$ on a manifold $\mathcal{M}_*$ with a low-LID point $x_0 \in \mathcal{M}_*$ (for example, set $\mathrm{LID}_*(x_0) = 0$). A perfect model will exhibit DD-Mem at $x_0$, but for any ball $B_r^d(x_0)$ containing $x_0$, $P_\theta(B_r^d(x_0)) = P_*(B_r^d(x_0))$, violating the second condition of Definition A.1. □

Since DD-Mem is a consequence of the data distribution having points $x_0$ with inherently low $\mathrm{LID}_*(x_0)$, these memorized points are likely to be generated even when there is no excess probability mass assigned near $x_0$ by $P_\theta$, as required in the definition of $(\lambda, \gamma)$-copies.

**Theorem A.3** (Informal)**.** *Suppose $x_0 \in \mathbb{R}^d$ is such that $p_\theta(x)$ exhibits OD-Mem at $x_0$. Then, for every $\lambda > 1$ and $0 < \gamma < 1$, $p_\theta(x)$ will generate $(\lambda, \gamma)$-copies of $x_0$ with near-certainty.*

*Proof.* See Appendix E for the formal statement of the theorem and proof. □

The MMH thus provides two important pieces of context for the definition of $(\lambda, \gamma)$-copies. The first is that Definition A.1 is in some sense incomplete; it does not cover DD-Mem. The second is that OD-Mem can be considered a useful refinement of Definition A.1. While the algorithm given by Bhattacharjee et al. (2023) is intractable at scale, we show in Section 4 that the added strength (in the mathematical sense) of the MMH allows us to flag memorized data more efficiently using only estimators of LID.

## B  LID ESTIMATION

### B.1  LID ESTIMATION WITH DIFFUSION MODELS

As mentioned in the main manuscript, we follow the SDE framework of Song et al. (2021) for DMs, where the so-called forward SDE is given by

$$\mathrm{d}x_t = f(x_t, t)\mathrm{d}t + g(t)\mathrm{d}W_t, \quad x_0 \sim p_*(x), \tag{3}$$

where $f : \mathbb{R}^d \times [0,1] \to \mathbb{R}^d$ and $g : [0,1] \to \mathbb{R}$ are pre-specified functions and $W_t$ is a Brownian motion on $\mathbb{R}^d$. This process progressively adds noise to data from $p_*(x)$, and we denote the distribution of $x_t$ as $p_t(x_t)$. This process can be reversed in time in the sense that if $y_t := x_{1-t}$, then $y_t$ obeys the so-called backward SDE,

$$\mathrm{d}y_t = \left[ g^2(1-t)\nabla \log p_{1-t}(y_t) - f(y_t, 1-t) \right]\mathrm{d}t + g(1-t)\mathrm{d}\tilde{W}_t, \quad y_0 \sim p_1, \tag{4}$$

where $\tilde{W}_t$ is another Brownian motion. DMs aim to learn the (Stein) score function, $\nabla \log p_t(x_t)$, by approximating it with a neural network $s_\theta : \mathbb{R}^d \times (0,1] \to \mathbb{R}^d$. Once the network is trained, $s_\theta(x_t, t) \approx \nabla \log p_t(x_t)$ is plugged in Equation 4, and solving the resulting SDE transforms noise into model samples. Below we briefly summarize two existing methods, FLIPD and NB, for approximating $\mathrm{LID}_\theta(x)$ for a DM.

**FLIPD**  Kamkari et al. (2024b) proposed FLIPD, an estimator of $\mathrm{LID}_\theta(x)$ for DMs. Commonly $f$ is linear in $x_t$, in which case the transition kernel corresponding to the forward SDE is given by

$$p_{t|0}(x_t \mid x_0) = \mathcal{N}(x_t; \psi(t)x_0, \sigma^2(t)I_d), \tag{5}$$

where $\psi, \sigma : [0,1] \to \mathbb{R}$ are known functions which depend on the choices of $f$ and $g$, and which can be easily evaluated. For a DM with such a transition kernel, FLIPD is defined as

$$\mathrm{FLIPD}(x, t_0) = d + \sigma^2(t_0)\Big( \mathrm{tr}\left( \nabla s_\theta\big(\psi(t_0)x, t_0\big) \right) + \|s_\theta\big(\psi(t_0)x, t_0\big)\|^2 \Big), \tag{6}$$

where $t_0 \in [0,1]$ is a hyperparameter. Kamkari et al. (2024b) proved that, when $t_0 \approx 0$ and $x \in \mathcal{M}_\theta$, $\mathrm{FLIPD}(x, t_0)$ is a valid approximation of $\mathrm{LID}_\theta(x)$. The reason for this is that the rate of change of the log density of the convolution between $p_\theta(x)$ and a Gaussian evaluated at $x_0$ with respect to the amount of added Gaussian noise approximates $\mathrm{LID}_\theta(x_0)$, and Kamkari et al. (2024b) showed that FLIPD computes this rate of change. In practice, computing the trace of the Jacobian of $s_\theta$ is the only expensive operation needed to compute FLIPD, and this is easily approximated by using the Hutchinson stochastic trace estimator (Hutchinson, 1989).

**NB**  Stanczuk et al. (2024) proposed another estimator of $\mathrm{LID}_\theta(x)$ for DMs. Following Kamkari et al. (2024b), we refer to this estimator as the normal bundle (NB) estimator. Stanczuk et al. (2024) proved that when $f(x_t, t) \equiv 0$, $s_\theta(x_t, t)$ points orthogonally towards $\mathcal{M}_\theta$ as $t \to 0$. They leverage this observation as follows: for a given $x$, Equation 3 is started at $x$ and run forward until time $t_0$; this is done $k$ times, resulting in $x_{t_0}^{(1)}, \dots, x_{t_0}^{(k)}$. The matrix $S_\theta(x, t_0) \in \mathbb{R}^{d \times k}$ is then constructed as

$$S_\theta(x, t_0) = \left[ s_\theta\left(x_{t_0}^{(1)}, t_0\right) \Big| \cdots \Big| s_\theta\left(x_{t_0}^{(k)}, t_0\right) \right], \tag{7}$$

and thanks to the previous observation, the columns of $S_\theta(x, t_0)$ approximately span the normal space of $\mathcal{M}_\theta$ at $x$ when $t_0 \approx 0$, meaning that $\mathrm{rank}\, S_\theta(x, t_0) \approx d - \mathrm{LID}_\theta(x)$. The NB estimator is given by

$$\mathrm{NB}(x, t_0) = d - \mathrm{rank}\, S_\theta(x, t_0). \tag{8}$$

In practice the rank is numerically computed by setting a threshold, carrying out a singular value decomposition of $S_\theta(x, t_0)$, and counting the number of singular values above the threshold. Stanczuk et al. (2024) recommend setting $k = 4d$, and we follow this recommendation. Computing the NB estimator is much more expensive than FLIPD, since $4d$ forward calls have to be made to construct $S_\theta(x, t_0)$, and then the singular value decomposition has a cost which is cubic in $d$. Finally, we point out that when $f$ is not identically equal to 0, the NB method can be easily adapted to still provide a valid approximation of $\mathrm{LID}_\theta(x)$ (Kamkari et al., 2024a).

We highlight that both FLIPD and NB were originally developed as estimators of $\mathrm{LID}_*(x)$ under the view that if the learned score function is a good approximation of the true score function, then

$\text{LID}_\theta(x) \approx \text{LID}_*(x)$. In our work, we see these methods as approximating $\text{LID}_\theta(x)$. Note that these views are not contradictory: when the DM properly approximates the true score function, it will indeed be the case that $\text{LID}_\theta(x) \approx \text{LID}_*(x)$; importantly though, when this approximation fails, we interpret $\text{FLIPD}(x, t_0)$ and $\text{NB}(x, t_0)$ as still providing a valid approximation of $\text{LID}_\theta(x)$ rather than a poor estimate of $\text{LID}_*(x)$.

## B.2 LOCAL PRINCIPAL COMPONENT ANALYSIS

Local PCA (Fukunaga & Olsen, 1971) offers a straightforward method for estimating the $\text{LID}_*$ of a datapoint by using linear local approximations to the data manifold. Given $x$, local PCA first identifies a set of nearby points in the dataset, representing a neighbourhood; this is typically done through a $k$-nearest neighbours algorithm. Next, the algorithm performs a principal component analysis (PCA) on this neighbourhood to get $(i)$ principal components and $(ii)$ explained variances for each component; the resulting principal components capture the directions of data variation, with the explained variance showing the amount of variation along each direction. Off-manifold directions are expected to have negligible explained variance. Hence, local PCA determines the number of components with non-zero (or non-negligible) explained variance as an estimate for $\text{LID}_*(x)$.

## B.3 LID ESTIMATION WITH GANS

We assume the GAN is given by a generator $G_\theta : \mathbb{R}^{d'} \to \mathbb{R}^d$ which transforms latent variables from a distribution in $\mathbb{R}^{d'}$ to the ambient space $\mathbb{R}^d$. For a generated sample $x = G_\theta(z)$, we estimate $\text{LID}_\theta(x)$ as the rank of the Jacobian of the generator, i.e. rank $\nabla G_\theta(z)$. As for the NB estimator with DMs, the rank is numerically computed by thresholding singular values. We highlight that this is a standard approach to estimate $\text{LID}_\theta$ in decoder-based DGMs (Horvat & Pfister, 2022; Kamkari et al., 2024a; Humayun et al., 2024).

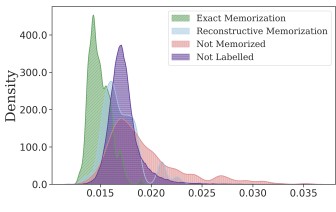 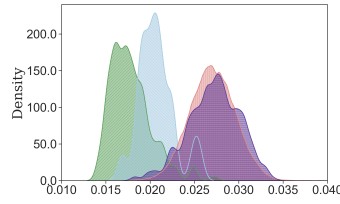 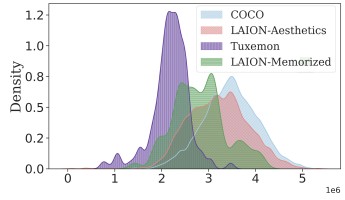

(a) $\widehat{\mathrm{LID}}_\theta$ adjusted by PNG for iDDPM.    (b) $\widehat{\mathrm{LID}}_\theta$ adjusted by PNG for StyleGAN-ADA2.    (c) PNG compression length for Stable Diffusion images.

Figure 8: Removing and analyzing image complexity as a confounding factor in memorization detection for CIFAR10 (a-b) and Stable Diffusion (c).

## C  EXPERIMENTAL DETAILS

### C.1  HYPER-PARAMETER SETUP FOR LID ESTIMATION METHODS

$\widehat{\mathrm{LID}}_*$ **with Local PCA**  As established in Appendix B.2, local PCA estimates the intrinsic dimensionality of a datapoint by counting the number of significant explained variances from PCA performed on the datapoint's local neighbourhood, determined by its $k$ nearest neighbours ($k = 100$ in our experiments). Finding the significant explained variances is done through a threshold hyper-parameter, $\tau$, where explained variances above $\tau$ are considered significant. For our approach in Figure 5d, we introduce two key modifications to better adapt the original Local PCA algorithm for detecting DD-Mem: $(i)$ instead of selecting $\tau$ individually for each datapoint, we define it globally as the 10th percentile of all explained variances across the entire dataset; $(ii)$ if a datapoint has neighbours within the 10th percentile of all pairwise distances, we restrict the neighbourhood to those points. The second modification allows us to avoid including distant points in the neighbourhood if closer ones already exist and especially helps us detect zero-dimensional point masses.

$\widehat{\mathrm{LID}}_\theta$ **for GANs**  As detailed in Appendix B.3, the rank of the Jacobian $\nabla G_\theta(z)$ can be used to estimate $\widehat{\mathrm{LID}}_\theta$. However, in practice, the rank — or, equivalently, the number of non-zero singular values — tends to equal the latent dimension; this is because singular values are typically close to zero but rarely exactly zero. To account for this, we apply a thresholding approach: a singular value is considered significant (non-zero) if it exceeds a hyperparameter $\tau$. We define $\tau$ as the 10th percentile of all singular values computed from the generated images in Figure 5b and Figure 8b.

$\widehat{\mathrm{LID}}_\theta$ **with NB**  We used $t_0 = 0.1$ and thresholded the singular values of $S_\theta(x, t_0)$ by 10th percentile; the results are presented in Figure 5c and Figure 8a. The choice of $t_0$ is empirically determined by observing how the NB score correlates with the memorization behavior with a fixed subset of 1000 randomly generated samples.

$\widehat{\mathrm{LID}}_\theta$ **with FLIPD**  Unless stated otherwise, we set $t_0 = 0.05$ for FLIPD and use the Hutchinson trace estimator to approximate the trace of the score gradient in Equation 6. In line with Kamkari et al. (2024b), we apply this in the latent space of Stable diffusion and use a single Hutchinson sample to estimate Equation 6 for all of our large-scale experiments.

**CFG Norm for Detecting Memorized Samples in the Training Set**  Note that while Wen et al. (2023) use the generation process to measure whether a synthesized image has been memorized, we were interested in detecting whether real, training-set images have been memorized in Figure 6, which requires some methodological changes. To compute a memorization score, we take $k$ Euler steps forward using the conditional score $s_\theta(x; t, c)$ with the probability flow ODE (Song et al., 2021) until time $t_0$ to get a point at $x_0 \in \mathbb{R}^d$. We then compute the CFG norm $\|s_\theta(x_0; t_0, c) - s_\theta(x; t_0, \emptyset)\|$. We use timestep $t_0 = 0.01$ and 3 Euler steps.

### C.2  THE CONFOUNDING EFFECT OF COMPLEXITY FOR DETECTING MEMORIZATION

$\mathrm{LID}_\theta$ is correlated with image complexity (Section 2; Kamkari et al. (2024b)), which raises a valid concern: the correlation, combined with the fact that simpler images are more likely to be memorized,

suggests that image complexity may confound our analysis. This is evident in Figure 5a (right panel), where GAN-generated images with the lowest $\widehat{\text{LID}}_\theta$ values are the simplest ones, not necessarily the memorized ones. To address this confounding factor, we draw inspiration from Kamkari et al. (2024b) and normalize it by PNG compression length, using it as a proxy for image complexity. We use the maximum compression level of 9 with the cv2 package (Bradski, 2000). According to this adjusted metric, the smallest values now correspond to memorized images that are not necessarily simple, such as the cars in CIFAR10. Figure 8a and Figure 8b show these adjusted LID estimated values, which achieve a slightly improved separation between memorized and not memorized images (as well as between exactly memorized and reconstructively memorized images) than the non-PNG-normalized results in the main text. It is worth noting that complexity did not appear to be a confounding factor in the Stable Diffusion analysis shown in Figure 6. In fact, as depicted in Figure 8c, the Tuxemon images are relatively simpler than the LAION memorized images, as measured by their PNG compression length. However, despite their simplicity, Tuxemon images have consistently higher $\widehat{\text{LID}}_\theta$ values compared to the memorized images in Figure 6b and Figure 6a.

# D  TEXT CONDITIONING AND MEMORIZATION IN STABLE DIFFUSION

## D.1  ADAPTING DENOISING DIFFUSION PROBABILISTIC AND IMPLICIT MODELS

Following Wen et al. (2023), we use denoising diffusion probabilistic models (DDPMs) (Ho et al., 2020). This model can be seen as a discretization of the forward SDE process of a score-based DM (Song et al., 2021). Here, instead of continuous timesteps $t$, a timestep $t$ instead belongs to a sequence $\{0, \ldots T\}$ with $T$ being the largest timescale; we use $T = 50$. We use the colour red to denote the discretized notation used by Ho et al. (2020).

With that in mind, DDPMs can be seen as a Markov noising process with the following transition kernel, parameterized by $\bar{\alpha}_t$, mirroring the notation of Ho et al. (2020):

$$p_{t|0}(x_t \mid x_0) \coloneqq \mathcal{N}(x_t; \sqrt{\bar{\alpha}_t} \cdot x_0, (1 - \bar{\alpha}_t)\mathbf{I}_d). \tag{9}$$

DDPMs do not directly parameterize the score function, but rather use a neural network $\epsilon_\theta(x_t, t)$, which relates to the score function as:

$$s_\theta(x, t/T) = -\epsilon_\theta(x, t)/\sqrt{1 - \bar{\alpha}_t}, \text{ or equivalently, } -\sigma(t/T)s_\theta(x, t/T) = \epsilon_\theta(x, t). \tag{10}$$

Note that in this context, we have $\sigma^2(t/T) = 1 - \bar{\alpha}_t$ and $\psi(t/T) = \sqrt{\bar{\alpha}_t}$. Equation 9 (the transition kernel) and Equation 10 (the score function) provide us with the recipe for estimating $\text{LID}_\theta$ using FLIPD with DDPMs (recall Equation 6).

When sampling from DMs we use the DDIM sampler (Song et al., 2020), mirroring the setup in Wen et al. (2023). In our notation, this sampler defines $\tilde{x}_t \coloneqq x_t/\psi(t)$, where $x_t$ is given as in Equation 3. In turn, $\tilde{x}_t$ obeys the forward SDE:

$$d\tilde{x}_t = \tilde{g}(t)dW_t, \quad \tilde{x}_0 \sim p_*(x), \tag{11}$$

where $\tilde{g}(t) = g(t)/\psi(t)$. This SDE has a corresponding score function

$$\tilde{s}_\theta(x, t) = \psi(t)s_\theta(\psi(t)x, t), \tag{12}$$

and DDIM uses this score function to sample from the model. The transition kernel corresponding to Equation 11 has $\tilde{\psi}(t) = 1$ and a $\tilde{\sigma}(t)$ which can be computed in closed form. Analogously to Equation 10, we can define $\tilde{\epsilon}_\theta$ as

$$\tilde{\epsilon}_\theta(x, t) = -\tilde{\sigma}(t/T)\tilde{s}_\theta(x, t/T). \tag{13}$$

We highlight that FLIPD (Equation 6) can be applied using the forward SDE in Equation 11 along with its corresponding score function in Equation 12, resulting in the estimate

$$\widetilde{\text{FLIPD}}(x, t) = d + \tilde{\sigma}^2(t)\Big(\text{tr}\big(\nabla\tilde{s}_\theta(x, t)\big) + \|\tilde{s}_\theta(x, t)\|^2\Big). \tag{14}$$

Note that this estimate can be computed when having access to $\tilde{\epsilon}_\theta$ thanks to Equation 13. We also note that in our text-conditioning analysis, we are interested in the probabilities conditioned by the text prompt, thus, these score functions are extended by the conditioning variable $c$, resulting in the modified forms $\epsilon_\theta(x; t, c)$, $\tilde{\epsilon}_\theta(x; t, c)$, $s_\theta(x; t/T, c)$, and $\tilde{s}_\theta(x; t/T, c)$.

## D.2  UNIFYING DIFFERENTIABLE METRICS FOR TEXT-CONDITIONED MEMORIZATION

We begin by revisiting the differentiable memorization metric used by Wen et al. (2023) for detecting and mitigating memorization, reformulating it within the continuous, score-based framework of diffusion models. Building on this, we perform an analysis, making minimal modifications to the original formulation to derive alternative metrics that remain effective and are theoretically-grounded. As a result, here we will formally derive three differentiable metrics: $\mathcal{A}^{\text{CFG}}(c)$, $\mathcal{A}^{s_\theta^{\text{CFG}}}(c)$, and finally $\mathcal{A}^{\text{FLIPD}}(c)$. We show that the value Wen et al. (2023) compute in their paper is in fact an estimator of $\mathcal{A}^{\text{CFG}}(c)$, rescaled by a constant. We then make minor modifications to introduce the two new metrics $\mathcal{A}^{s_\theta^{\text{CFG}}}(c)$ and $\mathcal{A}^{\text{FLIPD}}$ and interpret them through the lens of the MMH.

**The Differentiable Metric of Wen et al. (2023)**   For any text condition $c$, Wen et al. (2023) generate multiple samples $(\tilde{x}_0^{(n)})_{n=1}^N$, with the $n$th sample following the (DDIM) trajectory $\{\tilde{x}_T^{(n)}, \tilde{x}_{T-1}^{(n)}, \ldots, \tilde{x}_0^{(n)}\}$ from noise to data through the denoising process. They then introduce the following metric, which we have slightly reformulated to match our notation:

$$\mathcal{A}^{\mathrm{CFG}}(c; N, T) = \frac{1}{TN} \sum_{n=1}^N \sum_{t=0}^T \left\| \tilde{\epsilon}_\theta \left( \tilde{x}_t^{(n)}; t, c \right) - \tilde{\epsilon}_\theta \left( \tilde{x}_t^{(n)}; t, \emptyset \right) \right\|^2. \tag{15}$$

We colour-code the metric in red to distinguish between it and the analogous metric that we will shortly derive at the end of this section.

Let $\tilde{p}_t^{\mathrm{CFG}}$ represent the marginal probability at time $t$ induced by the DDIM sampler conditioned on $c$ with the addition of the CFG term. Recall from Equation 2 that the score used for sampling from $\tilde{p}_t^{\mathrm{CFG}}(\cdot \mid c)$ with CFG is

$$\tilde{s}_\theta^{\mathrm{CFG}}(x; t, c) = \tilde{s}_\theta(x; t, \emptyset) + \lambda(\tilde{s}_\theta(x; t, c) - \tilde{s}_\theta(x; t, \emptyset)). \tag{16}$$

Using Equation 13 and Equation 2, we can rewrite $\mathcal{A}^{\mathrm{CFG}}(c; N, T)$ as follows:

$$\mathcal{A}^{\mathrm{CFG}}(c; N, T) := \frac{1}{TN} \sum_{n=1}^N \sum_{t=0}^T \left\| -\frac{\tilde{\sigma}(t/T)}{\lambda} \left[ \tilde{s}_\theta^{\mathrm{CFG}}(\tilde{x}_t^{(n)}; t/T, c) - \tilde{s}_\theta(\tilde{x}_t^{(n)}; t/T, \emptyset) \right] \right\|^2. \tag{17}$$

We now assume $T \to \infty$, which will reformulate Equation 17 with an integral that we will replace with an expectation:

$$\mathcal{A}^{\mathrm{CFG}}(c; N) := \lim_{T \to \infty} \mathcal{A}^{\mathrm{CFG}}(c; N, T) \tag{18}$$

$$= \lambda^{-2} \cdot \frac{1}{N} \sum_{n=1}^N \int_0^1 \tilde{\sigma}^2(t) \| \tilde{s}_\theta^{\mathrm{CFG}}(\tilde{x}_t^{(n)}; t, c) - \tilde{s}_\theta(\tilde{x}_t^{(n)}; t, \emptyset) \|^2 \mathrm{d}t \tag{19}$$

$$= \lambda^{-2} \cdot \frac{1}{N} \sum_{n=1}^N \mathbb{E}_{t \sim \mathcal{U}(0,1)} \left[ \tilde{\sigma}^2(t) \| \tilde{s}_\theta^{\mathrm{CFG}}(\tilde{x}_t^{(n)}; t, c) - \tilde{s}_\theta(\tilde{x}_t^{(n)}; t, \emptyset) \|^2 \right] \tag{20}$$

$$= \lambda^{-2} \cdot \mathbb{E}_{t \sim \mathcal{U}(0,1)} \left[ \frac{1}{N} \sum_{n=1}^N \tilde{\sigma}^2(t) \cdot \| \tilde{s}_\theta^{\mathrm{CFG}}(\tilde{x}_t^{(n)}; t, c) - \tilde{s}_\theta(\tilde{x}_t^{(n)}; t, \emptyset) \|^2 \right]. \tag{21}$$

Here, $\mathcal{U}(0, 1)$ denotes the uniform distribution. Next, we observe that the inner term of the expectation on the right-hand-side of Equation 21 is in fact a Monte-Carlo estimator. By the law of large numbers, we have the following:

$$\mathcal{A}^{\mathrm{CFG}}(c) := \lim_{N \to \infty} \mathcal{A}^{\mathrm{CFG}}(c; N) \tag{22}$$

$$= \lambda^{-2} \cdot \mathbb{E}_{t \sim \mathcal{U}(0,1)} \mathbb{E}_{\tilde{x}_t \sim \tilde{p}_t^{\mathrm{CFG}}(\cdot | c)} \left[ \tilde{\sigma}^2(t) \cdot \| \tilde{s}_\theta^{\mathrm{CFG}}(\tilde{x}_t; t, c) - \tilde{s}_\theta(\tilde{x}_t; t, \emptyset) \|^2 \right]. \tag{23}$$

We now see that with the new formulation, all the red terms in Equation 23, have gone away, making it fully amenable to the score-based formulation of diffusion models. The $\lambda$ factor merely scales the metric, and for the purposes of detection and mitigation, this scaling is inconsequential: if a metric effectively predicts memorization, rescaling it will not diminish its effectiveness as a predictor. We thus disregard the scaling factor $\lambda$ to make the derivation cleaner and replace the uniform distribution $\mathcal{U}(0, 1)$ with a general "scheduling" distribution $\mathcal{T}(0, 1)$ of timesteps in $(0, 1]$; this would allow our metric to be a generalization of the one proposed by Wen et al. (2023):

$$\mathcal{A}^{\mathrm{CFG}}(c) := \mathbb{E}_{t \sim \mathcal{T}(0,1)} \mathbb{E}_{\tilde{x}_t \sim \tilde{p}_t^{\mathrm{CFG}}(\cdot | c)} \left[ \tilde{\sigma}^2(t) \cdot \| \tilde{s}_\theta^{\mathrm{CFG}}(\tilde{x}_t; t, c) - \tilde{s}_\theta(\tilde{x}_t; t, \emptyset) \|^2 \right]. \tag{24}$$

**Simplifying Further**   We have shown that the CFG vector norm and the CFG adjusted score norm behave similarly in Figure 3. If, instead of considering the CFG vector norm in Equation 24, we consider the CFG-adjusted score $\tilde{s}_\theta^{\mathrm{CFG}}(\cdot; t, c)$, we arrive at the following metric:

$$\mathcal{A}^{s_\theta^{\mathrm{CFG}}}(c) := \mathbb{E}_{t \sim \mathcal{T}(0,1)} \mathbb{E}_{\tilde{x}_t \sim \tilde{p}_t^{\mathrm{CFG}}(\cdot | c)} \left[ \tilde{\sigma}^2(t) \| \tilde{s}_\theta^{\mathrm{CFG}}(\tilde{x}_t; t, c) \|^2 \right]. \tag{25}$$

We have shown this to be a viable memorization metric, able to detect tokens driving memorization in Figure 11, and behaving comparable to $\mathcal{A}^{\mathrm{CFG}}(c)$, the original metric proposed by Wen et al. (2023). However, a nice property of $\mathcal{A}^{s_\theta^{\mathrm{CFG}}}(c)$ is that it can now be linked to MMH: for a memorized prompt where $\mathrm{LID}_\theta(\cdot \mid c)$ is small, the score function $\tilde{s}_\theta^{\mathrm{CFG}}(\cdot; t, c)$, especially for small $t$, tends to become large, causing the metric in Equation 25 to increase significantly.

**Linking to FLIPD** We now propose a more direct proxy for LID based on the FLIPD estimate of $\mathrm{LID}_\theta$. Recalling Equation 6, we can define the class-conditional $\mathrm{LID}_\theta(\cdot \mid c)$ estimate based on FLIPD as follows, analoguously to Equation 14:

$$\widetilde{\mathrm{FLIPD}}^{\mathrm{CFG}}(x;t,c) = d + \tilde{\sigma}^2(t) \cdot \left( \mathrm{tr}\big(\nabla \tilde{s}_\theta^{\mathrm{CFG}}(x;t,c)\big) + \|\tilde{s}_\theta^{\mathrm{CFG}}(x;t,c)\|^2 \right). \qquad (26)$$

Noting that $\widetilde{\mathrm{FLIPD}}^{\mathrm{CFG}}$ has a similar term to Equation 25, we add $d$ and the trace term from Equation 26 into Equation 25, and propose the following MMH-based metric:

$$d + \mathcal{A}^{s_\theta^{\mathrm{CFG}}}(c) + \mathbb{E}_{t\sim\mathcal{T}(0,1)}\mathbb{E}_{\tilde{x}_t\sim\tilde{p}_t^{\mathrm{CFG}}(\cdot|c)}\left[\tilde{\sigma}^2(t)\cdot\mathrm{tr}\left(\nabla\tilde{s}_\theta^{\mathrm{CFG}}(\tilde{x}_t;t,c)\right)\right] = \qquad (27)$$

$$\mathbb{E}_{t\sim\mathcal{T}(0,1)}\mathbb{E}_{\tilde{x}_t\sim\tilde{p}_t^{\mathrm{CFG}}(\cdot|c)}\left[\widetilde{\mathrm{FLIPD}}^{\mathrm{CFG}}(\tilde{x}_t;t,c)\right] =: \mathcal{A}^{\mathrm{FLIPD}}(c). \qquad (28)$$

Despite the fact that $\mathcal{A}^{\mathrm{FLIPD}}(c)$ can be expressed in terms of $\mathcal{A}^{s_\theta^{\mathrm{CFG}}}(c)$, the former indicates memorization when it is small, while the latter indicates memorization when it is large.

Note that while Equation 28 averages FLIPD values over (potentially) all the timesteps $t \in (0,1]$, the theory linking FLIPD and $\mathrm{LID}_\theta$ is only rigorously justified when $t \to 0$ (Kamkari et al., 2024b). Hence, we set the scheduling distribution $\mathcal{T}$ such that it primarily samples $t$ close to zero. As such, $\mathcal{A}^{\mathrm{FLIPD}}$ will average FLIPD estimate terms that are closely linked to $\mathrm{LID}_\theta(\cdot \mid c)$. Notably, our experiments also revealed that although setting $t$ as small as possible makes sense from a mathematical perspective, the score function, and as a result, FLIPD estimates, become unstable as $t \to 0$ (Pidstrigach, 2022; Kamkari et al., 2024b). Therefore, in practice, we choose $\mathcal{T}$ as a uniform supported on $(0.0, 0.2]$; therefore, putting more emphasis on these small $t$ values but at the same time avoiding instabilities in $\mathcal{A}^{\mathrm{FLIPD}}(c)$.

The scheduling is a small, but important distinction between $\mathcal{A}^{\mathrm{FLIPD}}$ on one hand, and $\mathcal{A}^{s_\theta^{\mathrm{CFG}}}$ and $\mathcal{A}^{\mathrm{CFG}}$ on the other hand; while $\mathcal{A}^{\mathrm{FLIPD}}$ sets $\mathcal{T}$ as a uniform on $(0.0, 0.2]$, $\mathcal{A}^{s_\theta^{\mathrm{CFG}}}$ and $\mathcal{A}^{\mathrm{CFG}}$ set $\mathcal{T}$ to a uniform distribution on $(0, 1]$, to mirror the setup in Wen et al. (2023).

### D.3 Increasing Image Complexity By Optimizing $\mathcal{A}^{\mathrm{FLIPD}}$

Wen et al. (2023) have an experiment where they optimize the prompt (embedding) $c$ directly to minimize $\mathcal{A}^{\mathrm{CFG}}(c)$, and as a result decrease $\mathcal{A}^{\mathrm{CFG}}(c)$, with the purpose of obviating memorization. Here, we take a similar approach but instead optimize $c$ to *maximize* $\mathcal{A}^{\mathrm{FLIPD}}(c)$.

In Figure 9, we optimize $c$ with Adam using multiple steps, and as we increase $\mathcal{A}^{\mathrm{FLIPD}}(c)$, we sample images using the prompt embedding which is being optimized. We see that images sampled from these prompts indeed increase in complexity. This is fully consistent with our expectations and understanding of LID. We see, however, that while at a certain range the images are relatively less memorized, the method tends to introduce excessively chaotic textures to artificially increase $\mathrm{LID}_\theta(\cdot \mid c)$, often at the expense of the image's semantic coherence. Despite this, we still find this to be an interesting result and invite future work on using different scheduling approaches for $\mathcal{A}^{\mathrm{FLIPD}}(c)$ that can stabilize the optimization process of $c$.

### D.4 Text Perturbation Approaches

**GPT-based Perturbations** As outlined in the main text, we sample $k$ tokens without replacement from a categorical distribution obtained by normalizing the token attributions, then use GPT-4 to replace these tokens; this ensures that tokens with the highest attributions are replaced more frequently. After selecting these $k$ tokens, we ask GPT to follow the instructions provided in Box 1 and use the output of the conversation as the new prompt. This process is repeated five times in our analysis to account for any randomness in the output of GPT. It is important to note that these perturbations are designed to preserve the semantic structure of the prompt. To ensure this, the instruction specifically asks GPT not to replace names of places or characters, and to keep the new prompt as semantically close to the original as possible.

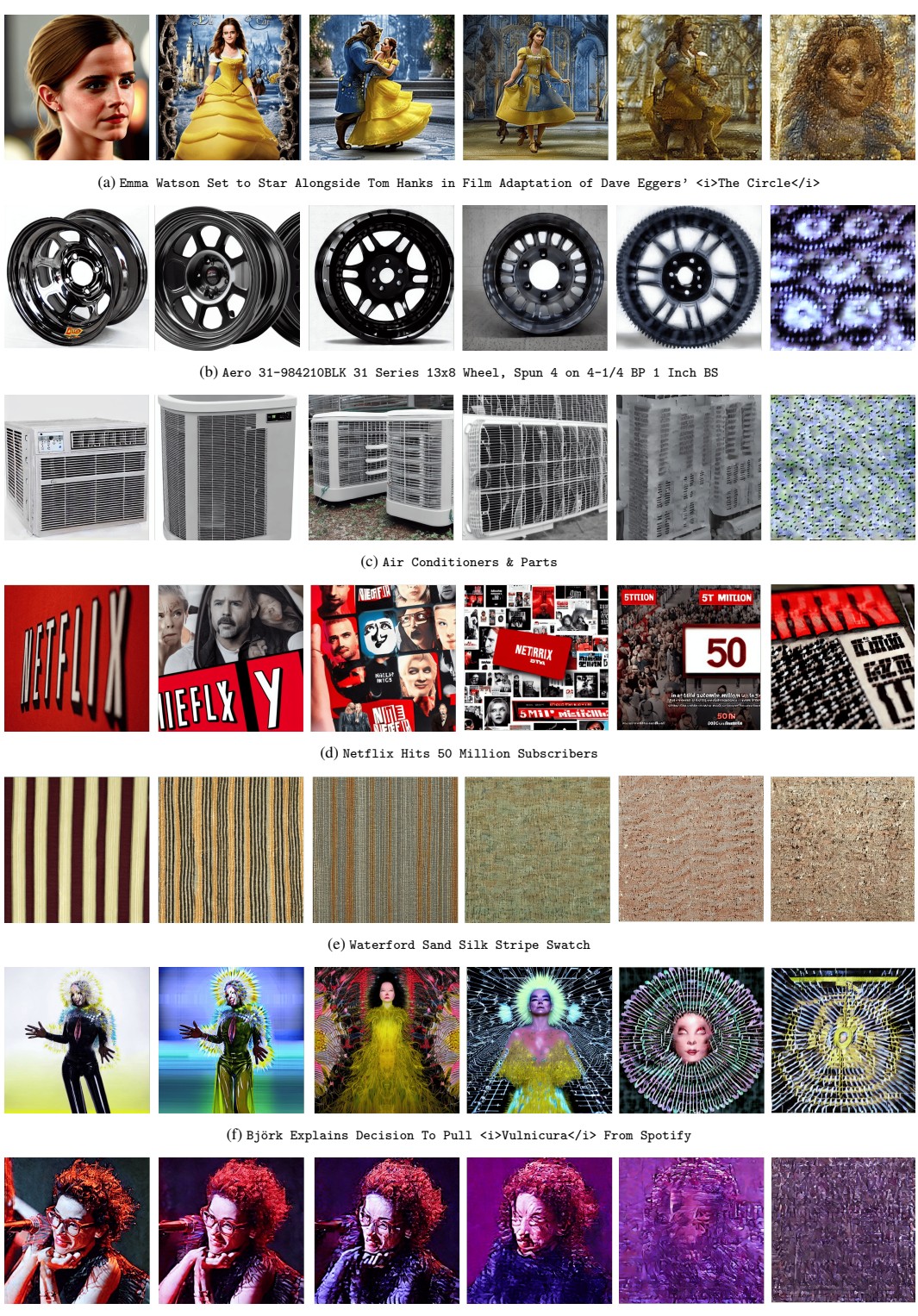

(a) `Emma Watson Set to Star Alongside Tom Hanks in Film Adaptation of Dave Eggers' The Circle`

(b) `Aero 31-984210BLK 31 Series 13x8 Wheel, Spun 4 on 4-1/4 BP 1 Inch BS`

(c) `Air Conditioners & Parts`

(d) `Netflix Hits 50 Million Subscribers`

(e) `Waterford Sand Silk Stripe Swatch`

(f) `Björk Explains Decision To Pull Vulnicura From Spotify`

(g) `Here's What You Need to Know About St. Vincent's Apple Music Radio Show`

Figure 9: Samples and their original memorized captions from directly optimizing text conditioning to increase $\mathcal{A}^{\text{FLIPD}}$ and reduce memorization. The images progress through different stages of the optimization process from left to right. While increasing $\text{LID}_\theta$ helps reduce memorization, uncontrolled increases often introduce chaotic textures, resulting in unrealistic images.

---

**Box 1: Instruction for perturbing a caption prompt using GPT-4**

I have the following caption as a sequence of tokens:
⟨ original_tokens ⟩

I want to create a new caption based on this one, but I want to perturb the following tokens:
⟨ selected_tokens ⟩

These are the rules to follow for perturbing tokens:

1. If the token is a special character or punctuation without significant semantics, you can remove it or change it to any special character

2. If the token is a number, you can replace it with another number that is close to it

3. If the token is a special name, such as the name of someone or some place or some culture, it should not be replaced

4. If the token is any other word, you can replace and rephrase it with any synonym that makes sense in the context

Given these requirements, please provide me with a new caption, not as a sequence of tokens, but as a natural language sentence that semantically matches closely with the original caption except for the perturbed tokens. Do not say anything else in response, only provide the new caption.

---

**Qualitative Comparison**   Figure 10 presents a qualitative comparison of our GPT-based perturbations applied to three memorized prompts. We have selected these specific examples to illustrate how the prompt perturbations function in practice. In this case, we set $k = 4$ and randomly perturb the tokens based on attributions derived from $\mathcal{A}^{\text{FLIPD}}$. Additionally, Figure 10 includes a column demonstrating the random token addition (RTA) approach proposed by Somepalli et al. (2023a), where $k$ random tokens from the CLIP library are inserted into the prompt. We see that the GPT-based perturbations better preserve the semantic integrity of the text caption, resulting in images that are not memorized and significantly more coherent.

| **Memorized Prompt** | **GPT + $\mathcal{A}^{\text{FLIPD}}$ Mitigation** | **RTA (Somepalli et al., 2023b)** |
|---|---|---|

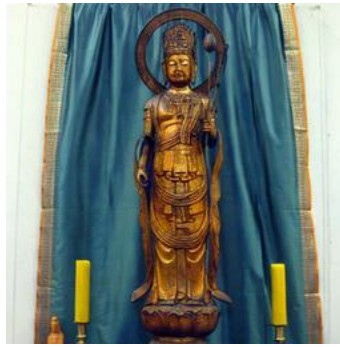 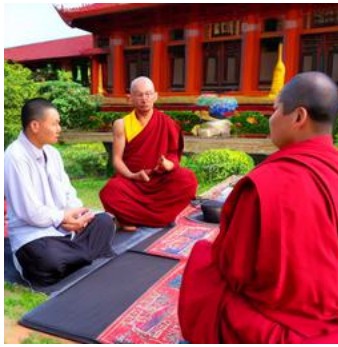 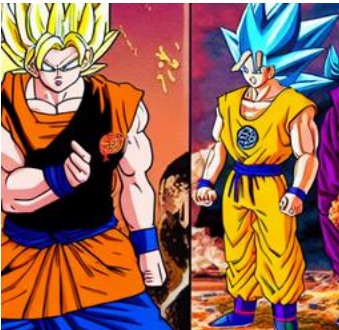

Talks on the Precepts and Buddhist Ethics

discussions about the precepts and Buddhist principles

Talks mellon dragonball on the villar Precepts and reformed Buddhist Ethics

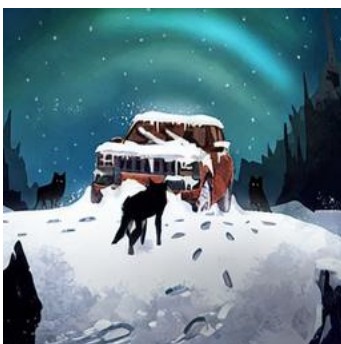 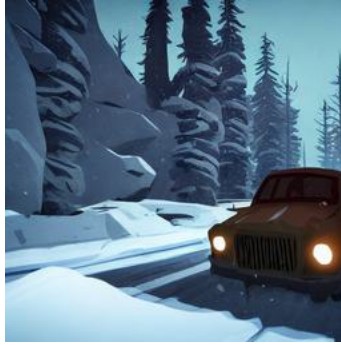 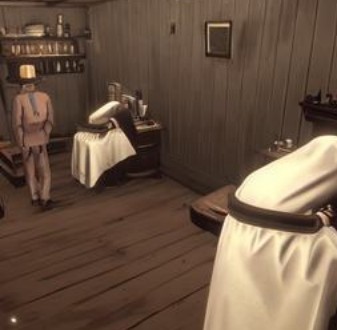

The Long Dark Gets First Trailer, Steam Early Access

The Long Dark Gets First Trailer; Steam Early Access

barbershop relying The idal Long Dark Gets First Trailer, Steam Early ghorn Access

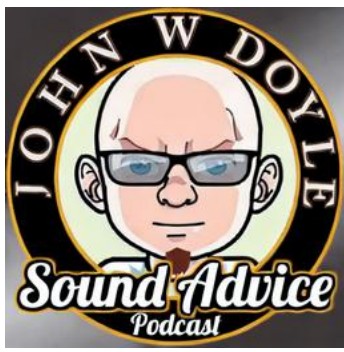 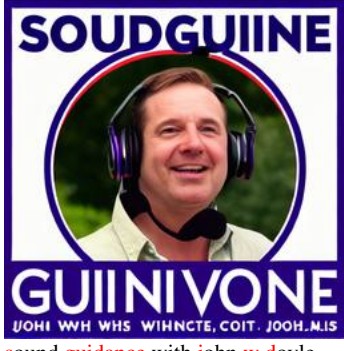 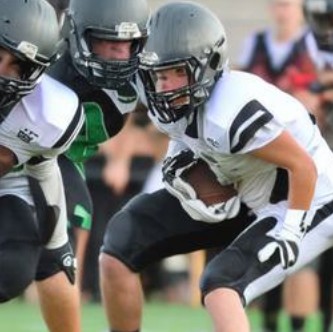

Sound Advice with John W Doyle

sound guidance with john w doyle

Sound Advice with John payments grill hsfb acadi W Doyle

Figure 10: Comparison of mitigation approaches. The tokens highlighted in red indicate the changes and perturbations made by each approach.

### D.5 MORE EXAMPLES OF TOKEN ATTRIBUTIONS

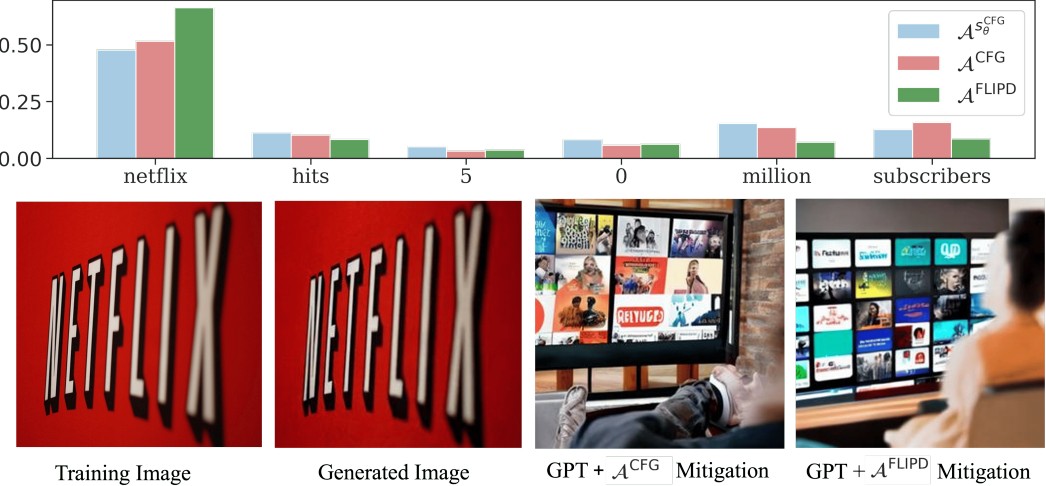

(a) All the methods detect "Netflix" a private trademark driving memorization.

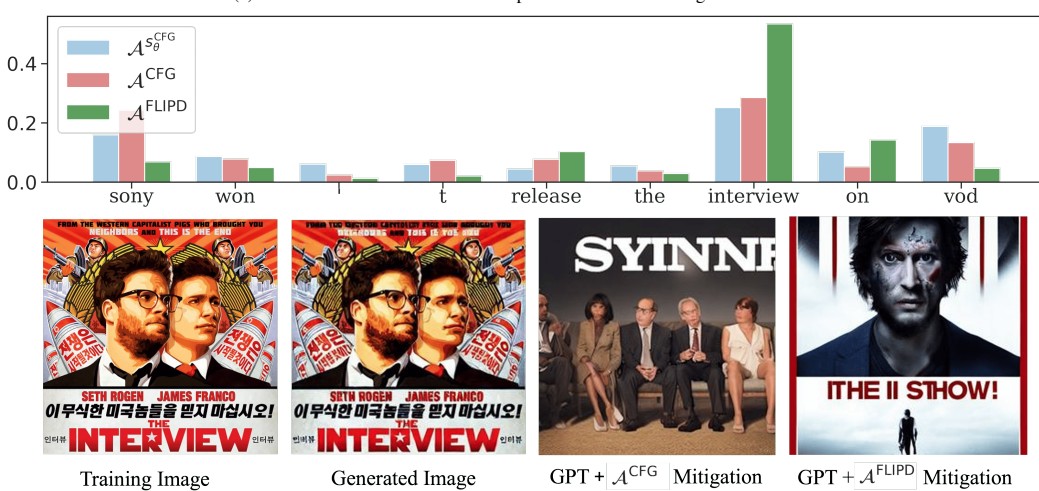

(b) All the methods detect "interview" (the movie title) as the driver for memorization, $\mathcal{A}^{\text{CFG}}$ also detects "Sony" as a significant token.

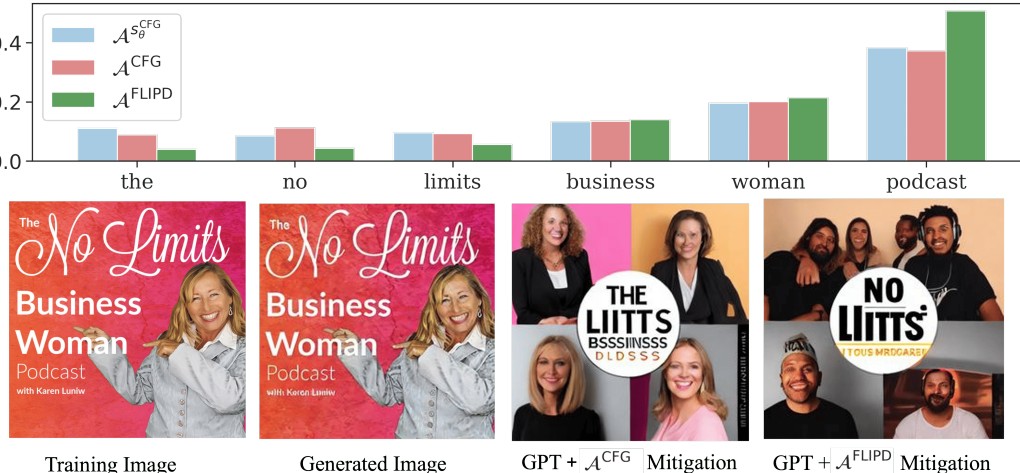

(c) All the methods detect "podcast" as the token driving memorization.

Figure 11: Memorized Stable Diffusion samples with a comparison of token attributions based on three different memorization metrics: the CFG vector norm proposed by Wen et al. (2023), the CFG-adjusted score $s_\theta^{\text{CFG}}(x; t, c)$ norm, and the FLIPD estimate for $\text{LID}_\theta(\cdot \mid c)$.

# E PROOFS

We restate each theorem in full formality below along with their proofs.

Throughout this section, we let $P_*$ and $P_\theta$ be the probability measures of the ground truth data and model, respectively. We assume that the respective supports of $P_*$ and $P_\theta$ are $\mathcal{M}_*, \mathcal{M}_\theta \subset \mathbb{R}^d$, smooth Riemannian submanifolds of the Euclidean space $\mathbb{R}^d$ with metrics $g_*$ and $g_\theta$ respectively. We denote the Riemannian measures on $\mathcal{M}_*$ and $\mathcal{M}_\theta$ as $\mu_*$ and $\mu_\theta$, respectively, so that $p_*(x) = \mathrm{d}P_*/\mathrm{d}\mu_*(x)$ and $p_\theta(x) = \mathrm{d}P_\theta/\mathrm{d}\mu_\theta(x)$. As mentioned in Section 2, we take a lax definition of manifold which allows them to vary in dimensionality in different components. A single manifold under our definition is equivalent to a disjoint union of manifolds under the more standard definition.

## E.1 PROPOSITION 3.1

**Lemma E.1.** *Assume that $p_*(x) > 0$ for every $x \in \mathcal{M}_*$, and let $x_0 \in \mathcal{M}_*$. Then, the following are equivalent:*

1. *$P_*(\{x_0\}) > 0$, and*

2. *$LID_*(x_0) = 0$.*

*Proof.*

$(1) \implies (2)$ Assume $P_*(\{x_0\}) > 0$.

$$0 < P_*(\{x_0\}) = \int_{\{x_0\}} p_*(x)\mathrm{d}\mu_*(x), \tag{29}$$

which necessitates $\mu_*(\{x_0\}) > 0$. If we had $\mathrm{LID}_*(x_0) > 0$ this would incur a contradiction: letting $(U, \phi)$ be a chart around $x_0$, then by the definition of $\mu_*$,

$$0 < \mu_*(\{x_0\}) = \int_{\phi(\{x_0\})} \sqrt{\det(g_*)}d\lambda, \tag{30}$$

where $\lambda$ is the Lebesgue measure on $\mathbb{R}^{\mathrm{LID}_*(x_0)}$ or the counting measure if $\mathrm{LID}_*(x_0) = 0$. Due to the singleton domain of integration, positivity of the integral in Equation 30 would be impossible *unless* $\mathrm{LID}_*(x_0) = 0$.

$(2) \implies (1)$ Suppose $\mathrm{LID}(x_0) = 0$. This implies that $\{x_0\}$ is an open set in the subspace topology of $\mathcal{M}_*$. Since $x_0 \in \mathrm{supp}\,\mu_*$, any open set containing $x_0$ must have positive measure under $\mu_*$, so that $\mu_*(\{x_0\}) > 0$. Then, since $P_*(\{x_0\}) = p_*(x_0)\mu_*(\{x_0\})$ and $p_*(x_0) > 0$, it follows that $P_*(\{x_0\}) > 0$.

$\square$

**Proposition E.2** (Formal Restatement of Proposition 3.1)**.** *Assume that $p_*(x) > 0$ for every $x \in \mathcal{M}_*$. Let $\{x_i\}_{i=1}^n$ be a training dataset drawn independently from $p_*(x)$. Then:*

1. *If duplicates occur in $\{x_i\}_{i=1}^n$ with positive probability, then they will occur at a point $x_0$ such that $LID_*(x_0) = 0$.*

2. *If $LID_*(x_0) = 0$ for some $x_0 \in \mathcal{M}_*$, then the probability of duplication in $\{x_i\}_{i=1}^n$ will converge to 1 as $n \to \infty$.*

*Proof.*

(1) Due to Lemma E.1, it suffices to show that any duplicates in $\{x_i\}_{i=1}^n$ must occur at a point $x_0$ such that $P_*(\{x_0\}) > 0$. It is thus enough to show that if $P_*(\{x_0\}) = 0$ for every $x_0 \in \mathcal{M}_*$, then $P_*(x_1 = x_2) = 0$. Assume that $P_*(\{x_0\}) = 0$ for every

$x_0 \in \mathcal{M}_*$. Since $x_1$ and $x_2$ are independent, $P_*(x_1 = x_2) = P_* \times P_*(D)$, where $D = \{(x, x) \in \mathcal{M}_* \times \mathcal{M}_* \mid x \in \mathcal{M}_*\}$. We then have:

$$P_* \times P_*(D) = \int_D \mathrm{d}P_* \times P_*(x_1, x_2) = \int_{\mathcal{M}_*} \int_{\{x_2\}} \mathrm{d}P_*(x_1) \mathrm{d}P_*(x_2) \tag{31}$$

$$= \int_{\mathcal{M}_*} P_*(\{x_2\}) \mathrm{d}P_*(x_2) = 0, \tag{32}$$

where the second equality follows from Fubini's theorem (see e.g. Theorem 7.26 in Folland (2013)), and the last equality follows by assumption. This finishes this part of the proof.

(2) Suppose $\mathrm{LID}_*(x_0) = 0$ for some $x_0 \in \mathcal{M}_*$. By Lemma E.1, we have $P_*(\{x_0\}) > 0$. In this case, $P_*(x_i = x_0) > 0$ for all $i \in \{1, \ldots, n\}$, meaning that

$$P_*(x_i = x_j \text{ for some } 1 \leq i < j \leq n) \geq P_*(x_i = x_j = x_0 \text{ for some } 1 \leq i < j \leq n) \tag{33}$$

$$\geq 1 - P_*(x_i \neq x_0 \text{ for all } i \geq 2) \tag{34}$$

$$= 1 - P_*(x_2 \neq x_0) \cdots P_*(x_n \neq x_0) \tag{35}$$

$$= 1 - (1 - P_*(\{x_0\}))^{n-1} \tag{36}$$

$$\longrightarrow 1, \tag{37}$$

where the last line depicts the limiting behaviour as $n \to \infty$.

$\square$

## E.2 PROPOSITION 3.2

Here we presume the joint distribution of model samples and $k$-dimensional conditioning inputs $(x, c) \in \mathbb{R}^{d+k}$ has support $S \subset \mathbb{R}^{d+k}$ such that $\{x : (x, c) \in S \text{ for some } c \in \mathbb{R}^k\} = \mathcal{M}_\theta$. We define the *conditional support* of $x$ given $c$ to be $S(c) = \{x : (x, c) \in S\}$.

**Proposition E.3** (Formal Restatement of Proposition 3.2). *Let $x_0 \in \mathcal{M}_\theta$ and $c \in \mathbb{R}^k$. Suppose that $S(c)$ is also a submanifold of $\mathbb{R}^d$ and denote its LID at $x_0$ by $LID_\theta(x_0 \mid c)$. We then have*

$$LID_\theta(x_0 \mid c) \leq LID_\theta(x_0). \tag{38}$$

*Proof.* If $S(c)$ is a submanifold of $\mathbb{R}^d$, then it is also a submanifold of $\mathcal{M}_\theta$. The inequality follows directly. $\square$

## E.3 THEOREM A.3

Here we show that OD-mem implies data-copying under the definition of Bhattacharjee et al. (2023).

**Lemma E.4.** *Suppose $(\mathcal{M}, g)$ is a $d_0$-dimensional smooth Riemannian submanifold of Euclidean space $\mathbb{R}^d$. Let $\mu$ be the Riemannian measure of $\mathcal{M}$. If $B_r^d(x_0)$ denotes the $d$-dimensional ball of radius $r$ centred at $x_0$ in $\mathbb{R}^d$, then there exist constants $C_1^{\mathcal{M}} > 0$ and $C_2^{\mathcal{M}} > 0$ not depending on $r$ such that for all small enough $r$:*

$$C_1^{\mathcal{M}} r^{d_0} \leq \mu\left(B_r^d(x_0) \cap \mathcal{M}\right) \leq C_2^{\mathcal{M}} r^{d_0}. \tag{39}$$

*Proof.* Without loss of generality, by rotating and translating, we assume $x_0 = 0 \in \mathbb{R}^d$ and that the tangent plane of $\mathcal{M}$ in $\mathbb{R}^d$ at $x_0$ is $\mathbb{R}^{d_0} \times \{0\}^{d-d_0}$.

As $\mathcal{M}$ is smooth, in a neighbourhood $U$ of $x_0 = 0$, $\mathcal{M}$ can be written of a graph of a function $u : U \subset \mathbb{R}^{d_0} \to \mathbb{R}^{d-d_0}$, such that $u(0) = 0$ and all first derivatives vanish at $0$. Then, for small enough $r > 0$, we have

$$B_r^d(x_0) \cap \mathcal{M} = \{(z, u(z)) \in \mathbb{R}^d \mid z \in \mathbb{R}^{d_0}, \|z\|^2 + \|u(z)\|^2 < r^2\}. \tag{40}$$

Let

$$G(r) = \{(z, u(z)) \in \mathbb{R}^d \mid z \in \mathbb{R}^{d_0}, \|z\| < r\} \tag{41}$$

be the graph of $u$ in the open $d_0$-ball $B_r^{d_0}(0)$, and

$$\overline{G(r)} = \{(z, u(z)) \in \mathbb{R}^d \mid z \in \mathbb{R}^{d_0}, \|z\| \leq r\} \tag{42}$$

be the graph of $u$ in the closed $d_0$-ball $\overline{B_r^{d_0}(0)}$. Note that both are defined when $u$ is defined, i.e. small enough $r$, and are subset of $\mathcal{M}$. Then it is clear that we have

$$B_r^d(x_0) \cap \mathcal{M} \subseteq G(r). \tag{43}$$

Now, consider the function $v(z) = \frac{\|u(z)\|}{\|z\|}$. Note that $v(z)$ is continuous everywhere in $B_r^{d_0}(0) \setminus \{0\}$. Since $u$ and its derivatives vanish at $z = 0$, from the definition, we have $\lim_{z \to 0} v(z) = 0$ as well. Thus $v(z)$ can be extended to a continuous function in $B_r^{d_0}(0)$. Fix $R > 0$, and let $K$ be the maximum of $v$ over $\overline{B_R^{d_0}}$. Then, if $\|z\| < ar$ where $a = \frac{1}{\sqrt{1+K^2}}$, we have

$$\|z\|^2 + \|u(z)\|^2 = \|z\|^2(1 + v(z)^2) \leq \|z\|^2(1 + K^2) < r^2. \tag{44}$$

This shows that $G(ar) \subseteq B_r^d(x_0) \cap \mathcal{M}$ for $0 < r < R$. Thus we have

$$\mu(G(ar)) \leq \mu(B_r^d(x_0) \cap \mathcal{M}) \leq \mu(G(r)). \tag{45}$$

Let $K_1$ and $K_2$ be the minimum and maximum of $\sqrt{\det g}$ over $\overline{B_r^{d_0}(0)}$, respectively. Then we have

$$\mu(G(r)) = \int_{B_r^{d_0}(0)} \sqrt{\det g}\, dz \leq \int_{B_r^{d_0}(0)} K_2 dz = K_2 V_{d_0} r^{d_0} \tag{46}$$

and

$$\mu(G(r)) = \int_{B_r^{d_0}(0)} \sqrt{\det g}\, dz \geq \int_{B_r^{d_0}(0)} K_1 dz = K_1 V_{d_0} r^{d_0}, \tag{47}$$

where $V_{d_0}$ is the Euclidean volume of the unit $d_0$-ball. Combining the above results, we have

$$K_1 V_{d_0} a^{d_0} r^{d_0} \leq \mu(G(ar)) \leq \mu(B_r^d(x_0) \cap \mathcal{M}) \leq \mu(G(r)) \leq K_2 V_{d_0} r^{d_0}, \tag{48}$$

which finishes the proof with $C_1^{\mathcal{M}} = K_1 V_{d_0} a^{d_0}$ and $C_2^{\mathcal{M}} = K_2 V_{d_0}$. $\qquad\square$

**Theorem E.5** (Formal Restatement of Theorem A.3). *Assume that $p_*(x)$ and $p_\theta(x)$ are continuous and that $p_\theta(x)$ is strictly positive. Let $x_0 \in \mathcal{M}_\theta \cap \mathcal{M}_*$ and let $p_\theta(x)$ be a model undergoing OD-mem at $x_0$, i.e. $0 \leq LID_\theta(x_0) < LID_*(x_0)$. Then for any $\lambda > 1$ and $0 < \gamma < 1$, there exists a radius $r_0$ such that any $x \in B_{r_0}^d(x_0)$ is a $(\lambda, \gamma)$-copy of $x_0$ according to Definition A.1, and if $\{x_j\}_{j=1}^m$ is generated independently from $p_\theta(x)$, then the probability of $(\lambda, \gamma)$-copying $x_0$ converges to 1 as $m \to \infty$.*

*Proof.* For an arbitrary value of $r > 0$, we have that

$$P_\theta(B_r^d(x_0)) \geq \mu_\theta(B_r^d(x_0) \cap \mathcal{M}_\theta) \inf_{x \in B_r^d(x_0) \cap \mathcal{M}_\theta} p_\theta(x) \tag{49}$$

and similarly,

$$P_*(B_r^d(x_0)) \leq \mu_*(B_r^d(x_0) \cap \mathcal{M}_*) \sup_{x \in B_r^d(x_0)) \cap \mathcal{M}_*} p_*(x). \tag{50}$$

Using Lemma E.4,

$$\frac{P_*(B_r^d(x_0))}{P_\theta(B_r^d(x_0))} \leq \frac{\mu_*(B_r^d(x_0) \cap \mathcal{M}_*)}{\mu_\theta(B_r^d(x_0) \cap \mathcal{M}_\theta)} \cdot \frac{\sup_{x \in B_r^d(x_0) \cap \mathcal{M}_*} p_*(x)}{\inf_{x \in B_r^d(x_0) \cap \mathcal{M}_\theta} p_\theta(x)} \tag{51}$$

$$\leq \frac{C_2^{\mathcal{M}_*}}{C_1^{\mathcal{M}_\theta}} r^{LID_*(x_0) - LID_\theta(x_0)} \frac{\sup_{x \in B_r^d(x_0) \cap \mathcal{M}_*} p_*(x)}{\inf_{x \in B_r^d(x_0) \cap \mathcal{M}_\theta} p_\theta(x)}. \tag{52}$$

Note that by continuity and positivity of $p_\theta(x)$, $\inf_{x \in B_r^d(x_0) \cap \mathcal{M}_\theta} p_\theta(x)$ is bounded away from 0 as $r \to 0$, and by continuity of $p_*(x)$, $\sup_{x \in B_r^d(x_0) \cap \mathcal{M}_*}$ is bounded. In turn, since by assumption $LID_*(x_0) > LID_\theta(x_0)$, Equation 52 converges to 0 as $r \to 0$. As a result, there exists some $r_0$ sufficiently small enough for both

$$\frac{P_*(B_{r_0}^d(x_0))}{P_\theta(B_{r_0}^d(x_0))} \leq \frac{1}{\lambda} \tag{53}$$

and

$$P_*(B_{r_0}^d(x_0)) \leq \gamma \tag{54}$$

to be true (the latter arising from the fact that $P_*(B_r^d(x_0)) \to 0$ as $r \to 0$, which follows from $P_*$ being absolutely continuous with respect to $\mu_*$ and $\mu_*$ not assigning positive measure to singletons because $\mathrm{LID}_*(x_0) > 0$).

Thus, any $x \in B_{r_0}^d(x_0)$ is a $(\lambda, \gamma)$-copy of $x_0$. Since $B_{r_0}^d(x_0) \cap \mathcal{M}_\theta$ contains an open set in the subspace topology of $\mathcal{M}_\theta$, it follows that $\mu_\theta(B_{r_0}^d(x_0) \cap \mathcal{M}_\theta) > 0$. Then, since $p_\theta(x)$ is strictly positive, $P_\theta(B_{r_0}^d(x_0)) > 0$, so that

$$P_\theta(x_j \text{ is a } (\lambda, \gamma)\text{-copy of } x_0 \text{ for some } 1 \leq j \leq m) \tag{55}$$
$$= 1 - P_\theta(x_j \text{ is not a } (\lambda, \gamma)\text{-copy of } x_0 \text{ for every } 1 \leq j \leq m) \tag{56}$$
$$= 1 - P_\theta(x_1 \text{ is not a } (\lambda, \gamma)\text{-copy of } x_0) \cdots P_\theta(x_m \text{ is not a } (\lambda, \gamma)\text{-copy of } x_0) \tag{57}$$
$$\geq 1 - (1 - P_\theta(B_{r_0}^d(x_0)))^m \tag{58}$$

converges to 1 as $m \to \infty$.

$\square$

## F MEMORIZED IMAGES FROM CIFAR-10

We generate 50,000 images from each model. Since StyleGAN2-ADA is class-conditioned, we generate an equal number of samples per class. We then retrieve nearest neighbors from the training dataset using both $(i)$ SSCD distance (Pizzi et al., 2022) and $(ii)$ calibrated $\ell_2$ distance (Carlini et al., 2023). We find that each metric produces very different results, so we combine results from both to maximize our probability of identifying as many memorized images as possible. Furthermore, both metrics produce many false negatives, so we follow a manual process to produce accurate labels. For StyleGAN2-ADA, we take the closest 250 neighbours according to each distance, and for iDDPM, we take the top 300, producing a set of just under 500 or 600 images to visually examine for each model.[5] We then label all of these instances as either not memorized, exactly memorized, or reconstructively memorized (Somepalli et al., 2023a). All other images are not labelled, and have a low chance of being memorized.

Here we show each generated image we identified (odd rows) along with its matched training image (even rows below) for iDDPM and StyleGAN2-ADA on CIFAR10. For StyleGAN2-ADA, we labelled no pairs as reconstructive under the calibrated $\ell_2$ distance.

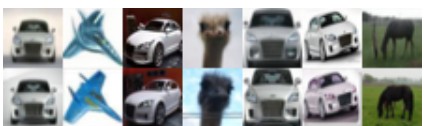

Figure 12: iDDPM: human-labelled reconstructive pairs in the top 300 according to calibrated $\ell_2$.

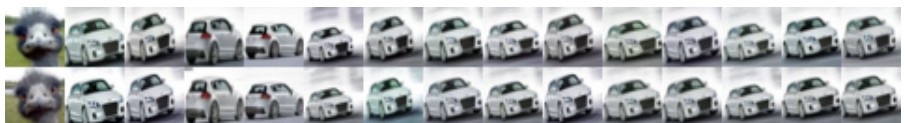

Figure 13: iDDPM: human-labelled exact pairs in the top 300 according to calibrated $\ell_2$.

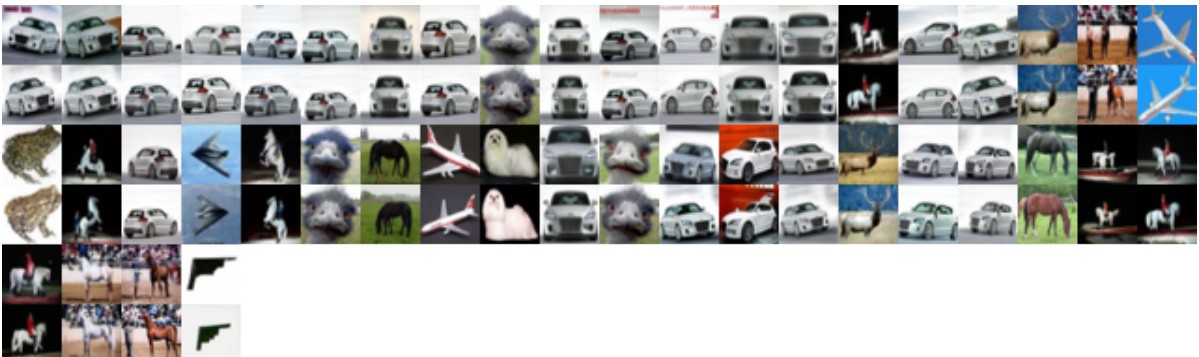

Figure 14: iDDPM: human-labelled reconstructive pairs in the top 300 according to SSCD.

---

[5]The cutoffs of 250 and 300 were chosen by inspection; from these ranks onwards, images ceased to appear memorized.

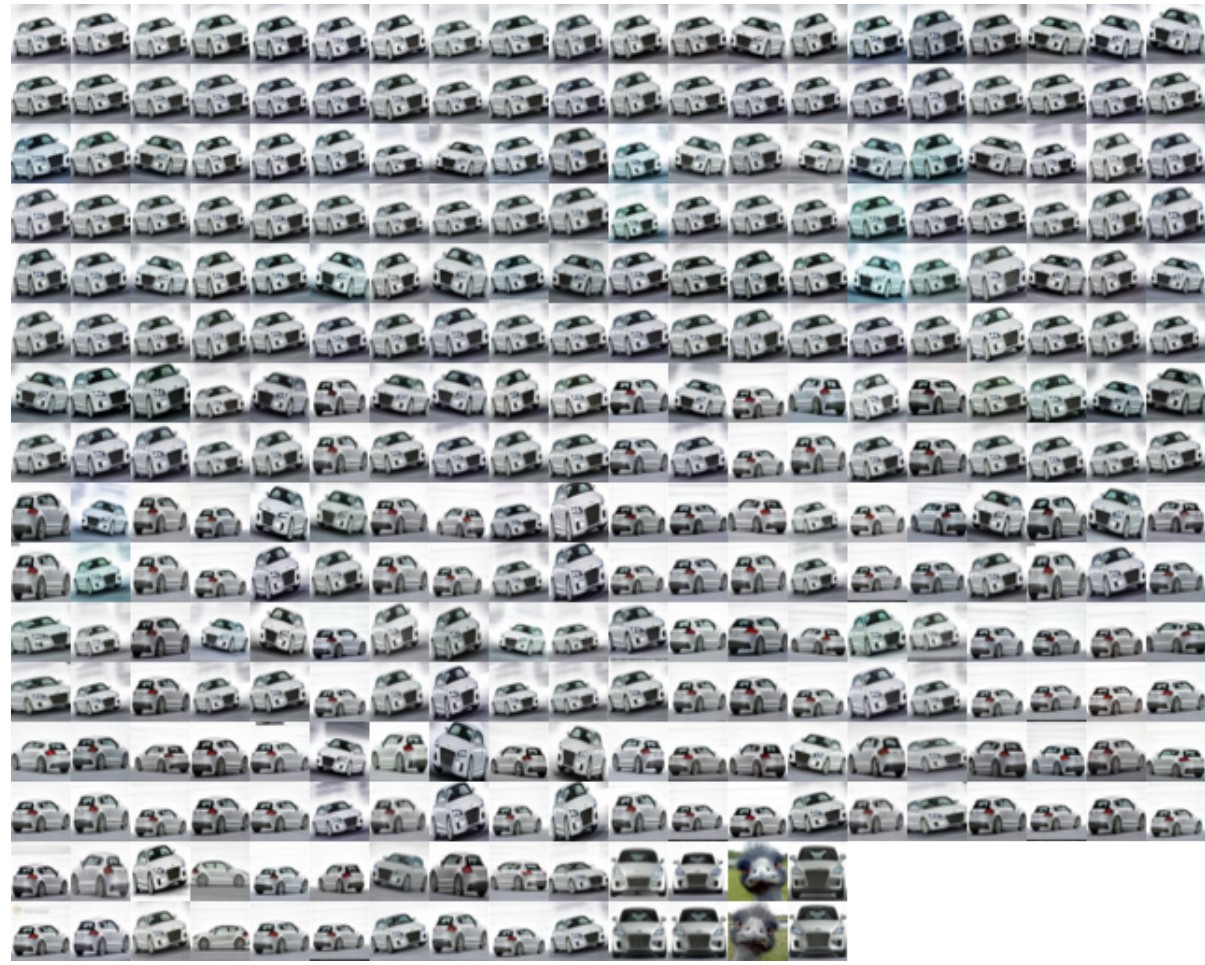

Figure 15: iDDPM: human-labelled exact pairs in the top 300 according to SSCD.

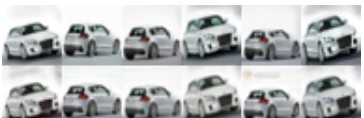

Figure 16: StyleGAN2-ADA: human-labelled exact pairs in the top 250 according to calibrated $\ell_2$.

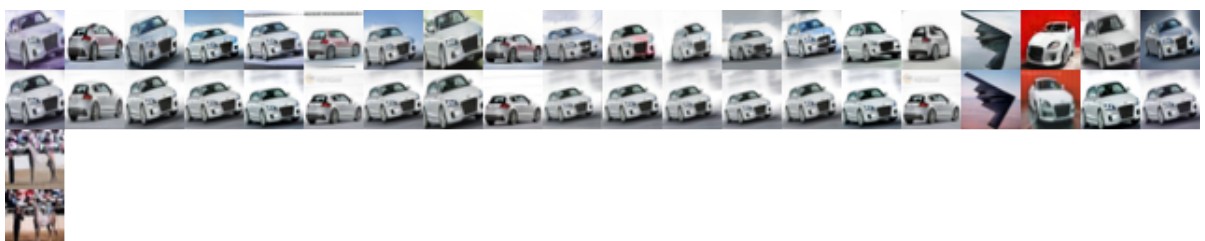

Figure 17: StyleGAN2-ADA: human-labelled reconstructive pairs in the top 250 according to SSCD.

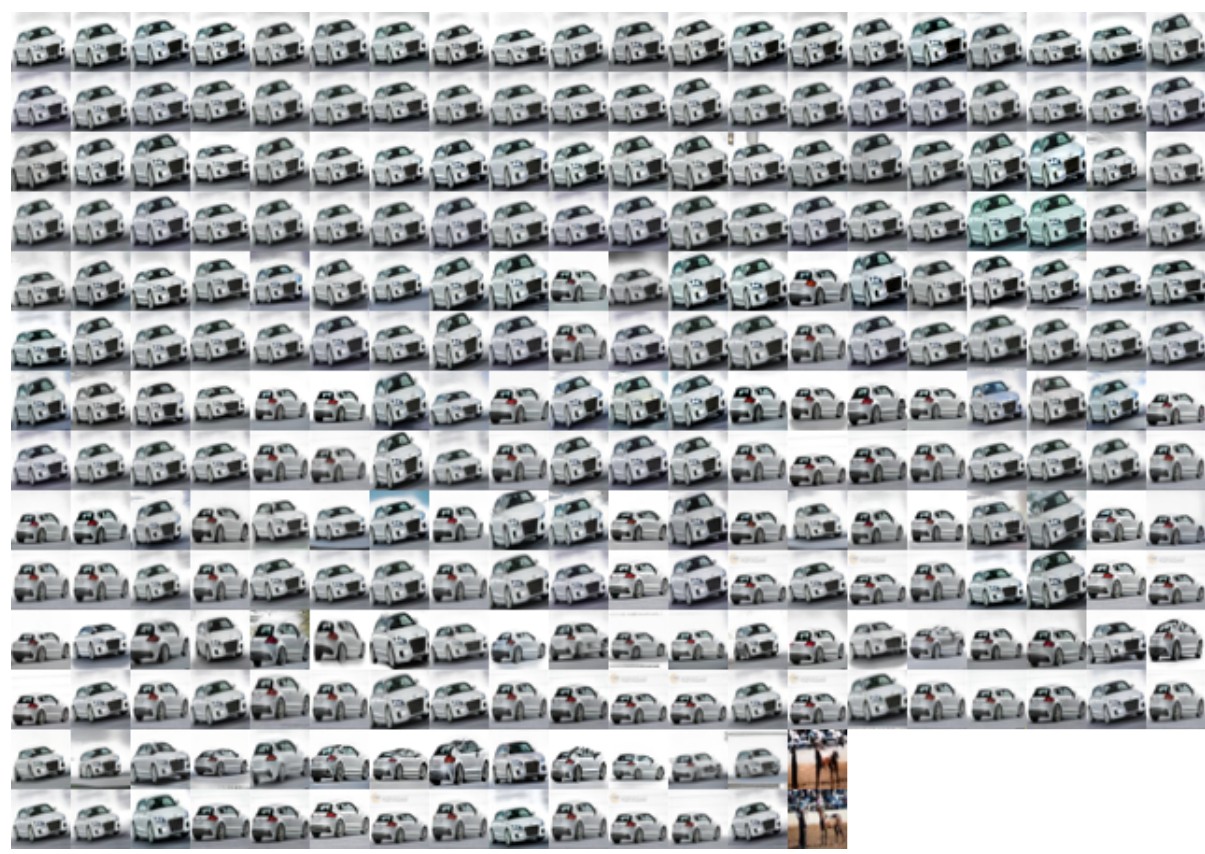

Figure 18: StyleGAN2-ADA: human-labelled exact pairs in the top 250 according to SSCD.

