# OpenReview forum: "A Geometric Framework for Understanding Memorization in Generative Models"
_ICLR.cc/2025/Conference — ICLR 2025 Spotlight_

### Official Review · Reviewer_bhfc · 2024-11-03

**Soundness:** 2
**Presentation:** 3
**Contribution:** 3
**Rating:** 6
**Confidence:** 3

**Summary:**

The paper proposes MMH, a geometric framework for understanding memorization in generative models by focusing on the dimensions of learned manifolds. It aims to analyze memorization as a function of the ground truth data manifold and the learned model manifold. The framework distinguishes between two types of memorization: overfitting-driven and data-driven, providing a theoretical base and empirical tools for detecting memorization across a range of models, from synthetic data to image generation using Stable Diffusion.

**Strengths:**

1. The authors combine the manifold memorization hypothesis and a geometric perspective to memorization, potentially unifying existing observations into a cohesive framework.
2. The paper provides clear motivation and background on memorization in DGMs, connecting memorization to legal and ethical implications.
3. The empirical experiments well support the propositions and theorems.

**Weaknesses:**

1. Some minor typos, for example ''subtantial'' should be ''substantial'' in line 44.
2. The MMH concept does not introduce substantial novel contributions beyond covering DD-Mem rather than only considering OD-Mem. This work is more like summarizing the existing work with their framework. For example, Proposition 3.5 is a (in)-formal summary of the findings in [1]'s experiments with MMH.

[1] Yuxin Wen, Yuchen Liu, Chen Chen, and Lingjuan Lyu. Detecting, explaining, and mitigating memorization in diffusion models. In Proc. ICLR, 2024.

**Questions:**

1. I am confused about the definition of LID. From the whole paper, I can tell the smaller the LID is the more memorization. In line 134, it is defined as ''the number of degrees of freedom'' instead of the number of variables needed in a minimal representation of the data. However, based on lines 125-126, it seems to be the same as the intrinsic dimension. Could you clarify?
2. In line 182: ''a notable consequence is that it cannot be detected by comparing training and test likelihoods''. Can you elaborate more about this?
3. As mentioned in the weaknesses, could you explicitly state your novel contributions more, or provide specific examples of how the framework goes beyond existing work? Currently, there is no comparison with the existing frameworks, such as [2]. I will be happy to increase my score if this is resolved.

[2] Robi Bhattacharjee, Sanjoy Dasgupta, and Kamalika Chaudhuri. Data-copying in generative models: a formal framework. In International Conference on Machine Learning, 2023.

---

> ### Author Response · Authors · 2024-11-16
> **Response to Reviewer bhfc**
>
> Thank you for your time, effort, and feedback! Thanks for pointing out the typo - we have addressed that and any other typos we could find in the manuscript. In response to your concerns:
>
> - Your main concern **(Weakness 2, Question 3)** was that our contributions were not clear enough. We appreciate this point, and have restructured the manuscript to make the contributions clearer (**descriptions of manuscript changes are bolded below**).
>
>   Indeed, the work’s contributions go well beyond DD-Mem. Our work is structured scientifically around a hypothesis: we propose it, show why it explains past observations, and then test it empirically. We then present a downstream application of our theory. Every step of this process is a contribution, and **we have updated our introduction to explicitly enumerate them. We have also relegated the former Section 3.1 connecting the MMH to definitions of memorization to the appendix because it is tangential to this story.**
>
>   The manifold memorization hypothesis is a fully novel lens from which to interpret memorization in DGMs. As a hypothesis, its value can be understood in two categories:
>   - **Explanatory:** The explanatory power of the MMH, while clear in hindsight, is based on several non-trivial connections we develop in Section 3. For example, your comment in Weakness 2, that Proposition 3.2 (3.5 in the former version) describes the results of Wen et al. (2023), is only clear after also seeing the connection between LID and the classifier-free guidance norm developed later in the section. We believe that altogether the explanatory power of this hypothesis and supporting theory constitute a substantial contribution over and above just summarizing existing work.
>   - **Practical:** To summarize the practical implications of the MMH, **we have added and reorganized some content into a series of paragraphs at the end of Section 2: “Why is the MMH Useful?” (L200-229)**. In comparison to past theoretical frameworks, it is useful because (1) its geometric perspective leads to computationally tractable tools (particularly in comparison to Bhattacharjee et al. (2023) (Question 3), which is not tractable at the scale of Stable Diffusion); (2) it is the only framework that also encompasses the “reconstructive memorization” of Figure 2, or in our terminology, memorization with $LID_\theta(x) > 0$; and (3) it distinguishes between OD-Mem and DD-Mem, leading to a differentiated understanding of the different types of memorization.
>
>   Finally, validating this hypothesis (Section 4.1) and devising inference-time mitigation approaches based on it (Section 4.2) are each further contributions of the paper.
>
> - **(Question 1)** To clarify, the earlier identity (L135-134 in the updated manuscript) is the definition: (A) $\text{LID}(x)$ is defined as the dimensionality of the manifold at $x$. All later descriptions of LID are meant to be heuristic interpretations of this definition. **We have reworded the manuscript to clarify this (L143).**
>
>   To address your specific question about the relationship between two further ways of describing LID:
>   - (B) The number of degrees of freedom at $x$.
>   - (C) The number of variables needed in a minimal representation of the data.
>
>   Both are common heuristic descriptions of LID, and as a first approximation, (A) = (B) ≈ (C). Locally, the local dimensionality of the manifold at $x$ (A) is equal to the number of degrees of freedom or the “number of independent dimensions you can move” along the manifold at $x$ (B), which is in turn equal to the minimal number of variables required to describe data on the manifold locally near $x$ (approximately C).
>
>   However, we don’t actually use the exact wording of (C) in our work, because if taken literally, it would imply some fixed number of variables required to encapsulate a global representation of the dataset. Datasets typically vary in complexity by region, so some regions need fewer variables to describe them (hence the word “local” in LID).

---

> > ### Author Response · Authors · 2024-11-16
> > **Response to Reviewer bhfc (cont'd)**
> >
> > - **(Question 2)** You asked about our assertion that DD-Mem (data-driven memorization) “cannot be detected by comparing training and test likelihoods.”
> >   - For context, OD-Mem (overfitting-driven memorization) on the other hand is overfitting, so it *can* be detected from the gap between training and test likelihoods. To be concrete, consider Figure 1(a) in our manuscript. The red model manifold is overly constrained around 3 of the training datapoints depicted on the left, but one could imagine a test datapoint from the blue ground truth manifold appearing outside of the red model manifold. These test datapoints outside of the model manifold would make test likelihood zero (or low, depending on how strictly the manifolds in the diagram are interpreted).
> >   - To finally answer your question, DD-Mem occurs when the model’s LID is low because the data manifold itself has low LID. This is not necessarily overfitting and can occur even when the model matches the ground truth density perfectly; in theory, the training and test likelihoods can even be equal in the presence of DD-Mem. As a result, comparing them gives no indication of DD-Mem.
> >
> > We hope these responses help to clarify your concerns and that our restructuring of the manuscript makes our contributions clearer; please let us know if you have any lingering concerns. We ask that you consider raising your score if you find them helpful.

---

> > > ### Comment · Reviewer_bhfc · 2024-11-18
> > >
> > > Thank you for providing detailed responses to my questions. The updated manuscript has effectively addressed my main concern regarding the contribution. I am pleased to increase my score in favor of acceptance.

---

> > > > ### Author Response · Authors · 2024-11-22
> > > > **Thank you!**
> > > >
> > > > We thank you again for the helpful feedback and for raising your score!

---

### Official Review · Reviewer_QHDy · 2024-11-04

**Soundness:** 4
**Presentation:** 2
**Contribution:** 3
**Rating:** 8
**Confidence:** 3

**Summary:**

Memorization is a big issue in deep generative models. Recent works have highlighted that such models are prone to duplicating their training points, given certain prompts, or generating samples that appear very close to training points with slight modifications (e.g., background and color). To combat and understand this phenomenon, this paper proposes a geometric framework which leverages the manifold hypothesis. Particularly, the work analyzes memorization in terms of the relationship between dimensionalities of the ground truth data manifold and the manifold that is learned by the model. This is done by analyzing and calculating the Local Instrinsic Dimension (LID) of the data and generated points, and building a theoretical understanding around the model's memorization behaviors. Overall, the paper validates the applicability of the manifold analysis in the detection of memorized samples through the development of new tools and prevention methods in generating such samples.

**Strengths:**

The paper provides a very valuable and interesting perspective about memorization, which comes in two forms: Overfitting-driven Memorization (OD-Mem) and Data-driven Memorization (DD-Mem). These two forms of memorization highlight that memorization is a convoluted aspect of trained deep networks, and it should not always be naively thought of as just overfitting. The paper does a decent job with making a case for this topic, by using LID as a metric of identifying memorized samples.

Furthermore, the work introduces a differential metric, $\mathcal{A}^\text{FLIPD}$, which allows for the optimization of a conditional variable $c$ to prevent the reproduction of training points during sampling. Specifically, the authors extended and combined the work of [Wen at al. (2024)](https://arxiv.org/pdf/2407.21720) and [Kamkari et al. (2024)](https://arxiv.org/pdf/2406.03537) to produce the aforementioned metrics, which show promising results in the prevention of memorization for stable diffusion models.

**Weaknesses:**

The comparison of various LID metrics to other memorization detection metrics is not quite clear, especially in figure 6. It is difficult to tell how much more effective LID metrics are since the authors did not contrast them well in their written text, and it is difficult to interpret the results in the density histograms of fig. 6 without showcasing what is classified as memorization, visually.

Furthermore, it is still unclear on how to quantitatively differentiate between OD-Mem and DD-Mem samples. The description, visualization (fig. 1), and toy example (fig. 4) certainly highlight the differences, but for natural datasets, it is difficult to tell still.

**Questions:**

For definition 3.1, could you elaborate on condition (2) or why $p_\theta$ cannot be equal to $p_*$?

Could you elaborate the differences between $s_\theta$ and $s_\theta^\text{CFG}$ in equation 41 of the Appendix D.2?

What is the conclusion of figure 6? The importance of this figure/result is not clear to me.

Could you provide more figures like figure (a), of memorized samples, in Figure 5? What was the threshold used in the detection metric from [Carlini et al. (2023)](https://arxiv.org/pdf/2301.13188), which was used to identify memorized samples?

Both $\hat{LID_\theta}$ and $\hat{LID_*}$ are estimators of LID. Is it fair to assume that $\hat{\text{LID}}_*$ is more accurate? Or, is that the wrong interpretation?

---

> ### Author Response · Authors · 2024-11-16
> **Response to Reviewer QHDy**
>
> Thank you for your detailed review and helpful feedback! We were especially thrilled by your comment that we provide an “interesting and valuable perspective about memorization.” We address your comments one by one below (apologies for the LaTeX rendering issues - we had to write some LaTeX in plaintext):
>
> - **(Weakness 1 and Question 3)** You expressed general concern about the clarity of Figure 6 and the comparison of LID metrics.
>   - First, we note that all metrics shown are either estimates of LID or proxies for LID; in particular, our analysis from the end of Section 3 predicts that the CFG vector norm varies inversely with LID. This connection was not explicit in Section 4 of our original manuscript; we thank you for raising this and have updated the manuscript to state this explicitly.
>   - Second, we note that the correct overall interpretation of the figure is a validation of the manifold memorization hypothesis: all estimators assign lower LIDs to memorized than non-memorized samples. We have slightly reworded part of this section to make this clearer.
>   - The memorized samples are those of Webster (2023), and some are available in Figure 11 in Appendix D.5 for reference.
>
> - **(Question 6)** You asked about comparing the accuracy of two estimators. First, it is important to point out that, while these are both indeed estimators of LID, they measure the LID of different things: `\widehat{LID}_*` estimates the LID of the ground truth manifold, while $\widehat{LID}_\theta$ estimates the LID of the model manifold. Since each estimates a different quantity and neither ground truth quantity is known for realistic datasets, it is difficult to authoritatively state which one is more accurate.
>
> - **(Weakness 2)** You mentioned that it is hard to quantitatively differentiate between OD-Mem and DD-Mem. We agree and believe this is a promising area for future work. There are a couple of practical avenues for differentiating these:
>   - If one can estimate both `\widehat{\text{LID}}_*` and `\widehat{\text{LID}}_\theta`, their relative values can be compared to differentiate between OD-Mem and DD-Mem. We do this for CIFAR-10 in Figure 5. Samples we label as exact memorization tend to have low values for both, which suggests that they are mostly undergoing DD-Mem. Unfortunately, $\widehat{\text{LID}}_*$ estimators do not scale well with dimensionality or dataset size (Tempczyk et al., 2022).
>   - In general, if memorization occurs for duplicates in the training data or overly simple images, this is suggestive of a low ground truth manifold dimension and therefore is likely DD-Mem.
>
> - **(Question 1)** You asked why it is necessary that $\lambda > 1$ in condition (2) of the definition of memorization by Bhattacharjee et al. (2023) (which we have now moved to Appendix A). The reason for this is that, if $\lambda = 1$, then the definition would label situation $P_\theta(B) = P_*(B)$ as memorization. This means the model could learn perfectly ($P_\theta = P_*$) and still be considered memorizing. While that could make sense in a DD-Mem scenario, we believe the original authors of this definition were aiming to capture overfitting behavior. (We remark that we have concluded that the former Section 3.1 on definitions of memorization is tangential to our thesis and have moved it to Appendix A.)
>
> - **(Question 2)** You asked about the difference between $s_\theta$ and $s_\theta^\text{CFG}$ in Appendix D.2. They are actually related by the definition of $s_\theta^\text{CFG}$ (Equation 2), which we have added to the appendix to clarify:
>
>   $$s_\theta^\text{CFG}(x; t, c) = s_\theta(x; t, \emptyset) + \lambda \big(s_\theta(x; t, c) - s_\theta(x; t, \emptyset)\big)$$
>
> - **(Question 4)** Thank you for asking about this - we are currently compiling the memorized examples. Once we have these next week, we will update the appendix with additional figures and provide a drive link of all our labelled generated and training sample pairs for each CIFAR-10 model along with a CSV containing the metrics for each. These will also be made available on GitHub when our code is released.
>
> - **(Question 5)** To clarify, we did not directly use a threshold on any quantitative metric to label memorized samples (L351-359). We found the two most common memorization metrics (SSCD of Pizzi et al. (2022) and calibrated L2 distance of Carlini et al. (2023)) were unreliable for our models and data. Instead, we used the method outlined in the “CIFAR10 Memorization” part of Section 4: with each metric, we filtered the top 300 “most memorized” images, and then labeled the top 300 pairs for each metric (600 total) by hand (as promised above, we will make all of these labels publicly available). (The corresponding thresholds before manual labeling were SSCD > 0.6854 or calibrated L2 < 0.8594 for iDDPM and SSCD > 0.6677 or calibrated L2 < 0.8750 for StyleGAN.)

---

> ### Comment · Reviewer_QHDy · 2024-11-17
>
> Thank you to the authors for taking their time to respond to my questions and comments. I am satisfied with their responses, which help clear up most of my misunderstandings. I am happy to say that I will raise the score of this paper to 8. To justify my decision, I would like to provide some clarifications.
>
> The main goal of this paper, to the best of my understanding, is to advocate for and connect manifold memorization hypothesis (MHN) to the understanding of memorization in deep generative models. I believe that the paper does demonstrate this aspect very well as the authors provide a very cool and new perspective on memorization, which separates it from the naive thinking that memorization is overfitting. This is a very wonderful perspective. Additionally, the paper also introduces two new metrics based on LID, which help prevent memorization in class-conditional diffusion models and show the potential usefulness of LID. Hence, the score of 8 is acceptable for me.
>
> Anyhow, ***I would like to continue a short discussion for my own understanding, which may help others***. On page 7, in the CIFAR10 memorization section, once you have generated the 50k samples. You utilized SSCD distance and l2-distance metrics from Carlini et al. (2023) to retrieve the most similar 600 images w.r.t the 50k set. What is the reason behind choosing the closest 300 images from both of the two metrics, respectively? Why not use just one of the two metrics?

---

> > ### Author Response · Authors · 2024-11-22
> > **Added images and discussion**
> >
> > We thank you for raising your score! We report that, as promised, we have added all the memorized images we labelled for CIFAR-10 and the full labelled dataset can be found in this anonymous [google drive folder](https://drive.google.com/drive/folders/1oJPls9woVtTpVbbqJYXFyUCqJC72MV29?usp=drive_link).
> >
> > Our reason for using both metrics was that, since neither metric is perfect, we wanted to maximize our probability of identifying as many memorized images as possible.

---

> ### Comment · Reviewer_QHDy · 2024-11-23
>
> Thank you very much. I suggest that you should add this reason/statement to your manuscript for clarification on the reason of using both metrics and to highlight their imperfection.
>
> Best of luck!

---

> > ### Author Response · Authors · 2024-11-25
> > **Thank you!**
> >
> > Thank you! We have incorporated your suggestion :)

---

### Official Review · Reviewer_6rGM · 2024-11-04

**Soundness:** 3
**Presentation:** 3
**Contribution:** 4
**Rating:** 8
**Confidence:** 3

**Summary:**

Memorization in Generative model occurs when model reproduce its training data instead generalising from it. This phenomenon is undesirable because it represent to modelling failure. The paper introduces the "manifold memorization hypothesis" (MMH) to explain memorization geometrically. It suggests that memorization happens when the manifold learned by the model at a point has a lower dimensionality than the true data manifold at that point. This framework defines two types of memorization:
Overfitting-Driven Memorization (OD-Mem): This occurs when the model assigns low dimensionality to a region of the manifold, overfitting specific training data points.
Data-Driven Memorization (DD-Mem): This happens when the true data distribution inherently has low dimensionality in certain regions, leading the model to reproduce specific details in these regions even if it generalizes well.

The paper introduces the Manifold Memorization Hypothesis (MMH) as a geometric framework to analyze and understand memorization in deep generative models (DGMs). It explores memorization through the lens of local intrinsic dimension (LID), a metric used to assess whether a model's manifold dimension aligns with that of the underlying data. The authors identify two types of memorization: overfitting-driven memorization (OD-Mem) and data-driven memorization (DD-Mem), categorizing instances of memorization based on LID discrepancies. This hypothesis is validated empirically with synthetic and real datasets, demonstrating how MMH can predict memorization tendencies and proposing methods to mitigate memorization in diffusion models, particularly Stable Diffusion.

**Strengths:**

The introduction of MMH offers a fresh perspective on memorization, connecting it to geometric properties and adding depth to existing theoretical understandings.
 The paper proposes actionable techniques to reduce memorization in generative models, which has direct implications for improving model safety and compliance.

**Weaknesses:**

The reliance on geometric and measure theory could make it challenging for a broader audience to fully understand and appreciate the framework.
There should be more experiments
While LID proves to be an effective measure, there is some overlap between memorized and non-memorized samples in LID estimates, which may affect its reliability in distinguishing memorized samples in real-world applications.

**Questions:**

How does the MMH framework extend to discrete generative models, such as those in NLP (transformer)?
Could alternative LID estimation methods reduce the overlap issue observed in the experiments, or is this an inherent limitation of the current approach?
Are there specific types of datasets or model architectures where MMH may not apply effectively?

---

> ### Author Response · Authors · 2024-11-16
> **Response to Reviewer 6rGM**
>
> Thank you for your positive review, and for your comment that the MMH “offers a fresh perspective on memorization, … and adding depth to existing theoretical understandings.” That was very much appreciated. To address the listed weaknesses and questions:
> - **(Weakness 1)** We agree with your comment that “the reliance on geometric and measure theory could make it challenging for a broader audience.” While this is partly unavoidable when proving theorems about it, the understanding of memorization expressed by the MMH is intuitive enough to be accessible to a broad ML audience (much like the manifold hypothesis itself).
>
>      To make this clearer in the paper, we have identified that the theory in the former Section 3.1 is mostly unnecessary for the core purpose of the paper and have therefore moved it to Appendix A. The section’s key takeaways have been distilled into discussion at the end of Section 2 (L200-229). Since this was probably the most technical section, we expect this will improve readability.
> - **(Weakness 2)** “There should be more experiments.” While more experiments are always better, we highlight that our comparison spans 4 models (including both diffusion and GANs) across 3 data regimes (with special focus on Stable Diffusion) with 5 proxies for LID and 3 differentiable scalars for inference-time mitigation. We believe this is sufficient to both validate our hypothesis and demonstrate its utility.
> - **(Weakness 3 and Question 2)** There is indeed some overlap between memorized and non-memorized samples in LID estimates. In our experience, no existing LID estimator is completely accurate, especially for high-dimensional, natural data which tends to be very noisy, so we anticipate that there is room to better apply the MMH via more precise LID estimates.
> - **(Question 1)** Any extension of the MMH to discrete models would be non-trivial - the manifold hypothesis and all the relevant theory do not immediately apply. Intuitively, because the manifold hypothesis expresses the “compressibility” of high-dimensional data, there may be an information theoretic analogy to LID that can be used to detect memorization, but we leave such an exploration for future work
> - **(Question 3)** In addition to LLMs, text, and discrete modelling in general, the MMH does not apply to any modelling regime that does not satisfy the manifold hypothesis, such as simple models that cannot learn manifold structure.

---

### Meta-Review · Area_Chair_5cYp · 2024-12-20

**Metareview:**

Several aspects of the work stand out. First, it offers actionable insights, presenting methods to mitigate memorization in diffusion models like Stable Diffusion. The empirical results substantiate the theoretical claims, highlighting the framework's utility across synthetic and real datasets. Second, it challenges conventional notions of memorization as mere overfitting, providing a more nuanced understanding that could shape future research in model safety and generalization.

The proposed LID-based metrics are particularly promising, enabling the detection of memorized samples and supporting practical advancements in generative modeling. Furthermore, the exploration of memorization’s geometric properties addresses ethical concerns around data leakage, showcasing the framework's relevance beyond academic boundaries. The work has received positive acknowledgment for these contributions, with multiple reviewers noting its intellectual depth, practical implications, and well-supported methodology.

Overall, the paper offers a fresh, impactful approach to a pressing problem, making meaningful theoretical and practical contributions to the study of generative models.

**Additional Comments On Reviewer Discussion:**

Good discussion, reviewers were happy with the discussion as all major issues were resolved.

---

### Decision · Program_Chairs · 2025-01-22

Accept (Spotlight)